

A localized plant species-specific BVOC emission rate library of
China established using a developed statistical approach based
on field measurements
Huijuan Han[1], Yanqi Jia[1], Rende Shi[2], Changliang Nie[1], Yoshizumi Kajii[1], Yan Wu[3],
Lingyu Li[1]*
[1]College of Environment and Geography, Carbon Neutrality and Eco-Environmental
Technology Innovation Center of Qingdao, Qingdao University, Qingdao 266071, China
[2]Eco-environment Monitoring Center of Qingdao, Shandong Province, Qingdao 266003,
China
[3]School of Environmental Science and Engineering, Shandong University, Qingdao 266237,
China
*Corresponding author. E-mail: lilingyu@qdu.edu.cn;
Contributing authors: hanhuijuan2022@163.com; jiayanqi2022@163.com;
yoyomjh@126.com; changliangnie@163.com; kajii.yoshizumi.7e@kyoto-u.jp;
wuyan@sdu.edu.cn

## Abstract

Precise quantification of biogenic volatile organic compound (BVOC) emissions is
essential for effective control of ozone and aerosol pollution. However, the lack of localized
and elaborate plant species-specific emission rate library pose significant challenges to its
accurate emission estimates in China. Also, large uncertainty exited in the representative
emission rate used in inventory compiling. Here a statistical approach of classifying emission
intensity and assigning the representative emission rates in higher accuracy was developed,
based on our and reported local field measurements. Furtherly, a localized plant
species-specific BVOC emission rate library for China for 599 plant species was established.
Critically, different reliability was given to each emission rate according to the measurement
technique. Emission simulations were made to evaluate the performance of the developed





library. By comparing with formaldehyde vertical column density observations, using our
localized library improved the model performance in catching the spatial variations of
isoprene emission. The newly estimated BVOC emissions were 27.70 Tg, 18% higher than
the estimations using the global library. The underestimates in the south and overestimates in
the northeast and west could be abated by updating the localized emission rates. More local
emission observations in higher reliability are encouraged to improve the accuracy of
emission rates and further the emission estimates.
**Keywords:** BVOC emission; Emission rate library; Localization; Enclosure measurement;
Emission inventory

## 1. Introduction

Biogenic volatile organic compounds (BVOCs) are primarily emitted by the vegetation
of terrestrial ecosystems (Ciccioli et al., 2023; Guenther et al., 1995, 2012; Li et al., 2023,
2024; Simpson et al., 1999). The compounds are highly reactive (Vella et al., 2023), which
can react with nitrogen oxides ($NO_x$) to generate ozone ($O_3$) and secondary organic aerosol
(SOA) through atmospheric oxidation (Li et al., 2022; Wei et al., 2024), thereby affecting air
quality, cloud formation, solar radiation transmission, and climate change (Blichner et al.,
2024; Ndah et al., 2024). Furthermore, the $O_3$ generation can be more sensitive to BVOCs
than $NO_x$ in VOCs-limited areas (Guo et al., 2024; Huang et al., 2024; Wang et al., 2023a). In
China, both the reduction of anthropogenic VOCs and growth of BVOC emissions in recent
years (Cao et al., 2022; Gai et al., 2024) made higher contributions of BVOCs to $O_3$ and SOA
formation (Yang et al., 2023). Cao et al. (2022) reported that the summertime BVOC
emissions led to an average increase of 8.6 ppb (17%) in daily maximum 8 h (MDA8) $O_3$
concentration and 0.84 μg m$^{-3}$ (73%) in SOA over China. Accurately estimating BVOC
emissions becomes essential for precise control of air pollution complex in China.
The reported BVOC emission inventories for China presented various results and large
uncertainty (Li et al., 2024). The quality of emission rates applied in the inventory will
significantly influence the accuracy of emission estimates (Wang et al., 2023b). In the existing



inventories, different emission rates were applied for the same plant species due to the various
assignment methods. Mostly the global emission rates by plant function type (PFT) were used,
which includes few observations in China. Differences usually exist in the emission of
domestic and foreign plants because of various genes and environment and climate (Chatani
et al., 2018). Errors will be definitely introduced when the foreign measurements are applied
in the Chinese emission inventory. Therefore, it is essential to localize the emission rate
library. Some inventories used partial local observations, but large uncertainty still existed.
The emission rate was assigned by one observed result or directly averaging several
observations. This assignment will cause errors because there are limited local measurements
and different studies may report various emission rates for the same plant species (Chen et al.,
2024; Zeng et al., 2024). Progressively, some researches applied a method of emission
intensity categories to determine the emission rates used in the inventory compilation
(Klinger et al., 2002; Wang et al., 2007; Yan et al., 2005). In this method, different emission
intensity categories (e.g., negligible, low, moderate, and high) are defined, with a
representative emission rate and a range of its ±50% for each category (Simpson et al., 1999;
Zhang et al., 2024). For each plant species, the emission rate is determined based on the
tendency of the reported emission rates to fall within certain categories. This method can
improve the accuracy of the final emission rates to a large extent; however, it has several
limitations. Firstly, the process of determining emission categories, representative emission
rates, and ranges is not straightforward and lacks theoretical evidence. Secondly, the various
emission categories and representative values in different studies led to disparate emission
rates for the same plant species. For example, Klinger et al. (2002) assigned the emission
rates of isoprene for *Saliix character, Quercus mongolica,* and *Picea jezoensis* as 70, 70, and
14 $\mu$g C g$^{-1}$ h$^{-1}$, while Wang et al. (2007) assigned them as 20, 50, and 10 $\mu$g C g$^{-1}$ h$^{-1}$. Thirdly,
most studies used coarse classifications of emission, typically five to seven categories
(Klinger et al., 2002; Yan et al., 2005), which might result in imprecise classifications and
overestimation or underestimation of emission rates for a specific plant species. Significant
uncertainties will be furtherly introduced into the BVOC emission estimates. Thus, it is



essential to have detailed emission categories and accurate representative values and ranges to
estimate emission rates accurately. Also, a localized BVOC emission rate library should be
established based on domestic observations to enhance the accuracy of inventory. Additionally,
the PFT averaged emission rates were usually used, failing to capture the species-specificity
of BVOC emissions. Research showed that isoprene emission rate could vary by 220–330%
among subtypes of vegetation. Therefore, it is also necessary to establish a plant
species-specific emission rate library.
In this study, we firstly conducted the emission measurements for plants in China to
provide more basal data for the establishment of localized emission rate library. Secondly, by
summarizing our field measurements along with the reported emission rates from China, a
statistical approach to determine the plant species-specific emission rate was developed. A
localized BVOC emission rate library of China was established and its features were
discussed. The differences in BVOC emission rates among different vegetation types, families,
genera, and species were explored. Then, the developed emission rate library was applied to
establish BVOC emission inventory for China using the Model of Emissions of Gases and
Aerosols from Nature (MEGAN) v3.2. Its performance was furtherly evaluated. Furthermore,
the influence of emission rate with different level of reliability on the estimated BVOC
emission was investigated. This study will be significant for improving the accuracy of local
biogenic emission inventory and furtherly the modeling of air quality. Also, our developed
statistical approach can be extended to the establishment of BVOC emission rate library for
other regions.

## 2. Field measurements of emission rates

Field measurements on BVOC emission rates were conducted from July 2020 to
September 2023. The sites covered the south and north of China, including Shandong, Hebei,
Jiangsu, and Anhui provinces. Their specific locations are shown in Figure S1. Meanwhile,
some pot experiments in the plant growth chamber were included. Totally, emissions from 66
plant species including 30 broadleaf tree, 12 coniferous tree, 20 shrub, two crop, and two herb




species were measured (Table S1). Dynamic enclosure technique was used for the
observations, as depicted in Figure S2 (Zhang et al., 2024). Firstly, selected branches were
enclosed within a Teflon bag with a volume ranging from 15 to 60 L (Welch Fluorocarbon,
Inc., USA) and PAR transparency close to 100%, which was passivated to avoid the
generation and adsorption of VOCs as much as possible. The clean air was continuously
introduced into the bag at a constant flow rate of 10–20 L min$^{-1}$ after removing water, $O_3$, and
VOCs through silicone rubber, potassium iodide and activated carbon. After equilibrium, the
gases in the bag were collected into adsorption tubes filled with Tenax TA and Carbograph
5TD (Markes International, Bridgend, UK) using an air-sampling pump (Gilian Gilair Plus,
Sensidyne, USA) with a flow rate of 200 mL min$^{-1}$ for 30 minutes. For each plant species,
three mature and healthy individuals were selected as replicates and one blank samples as the
background. During the whole enclosure, the temperature and photosynthetically active
radiation (PAR) were recorded in real time. After the experiment, all leaves on the enclosed
branch were collected and weighted after drying at 75 ℃ for 48 hours.

The sampled tubes were analyzed using the thermal desorption (TD) - gas

chromatography (GC) - mass spectrometry (MS) (TD, ATD II-26, Acrichi Inc., China;
GC-MS, 7890A-5975C, Agilent Technologies, USA). The detailed information about their
operation conditions can be referred to our previous study (Zhang et al., 2023, 2024). The
Agilent DB-5 chromatography column (30 m in height, inner diameter 0.25 mm, pore size
0.25 μm) was used. In the study, terpene mixed standards (Apelriemer Environmental, USA)
and photochemical assessment monitoring station (PAMS) mixed standards (LINDE, USA)
with a concentration of 1 ppm were used to quantify the concentration of VOCs. During the
quantification for one compound, the response factor (RF) method was used when the relative
standard deviation of RFs was < 20%. Otherwise, external standard method was used, and the
correlation coefficients of their curves were > 0.99. The quantified compounds included
isoprene, 14 monoterpenes, 6 sesquiterpenes, 21 alkanes, 4 alkenes, and 17 aromatics, listed
at Table S2.

The emission rates for each compound (VOC$_i$) were calculated as Equation (1).



$EF_i = \frac{F \times C_i}{M}$      (1)
where F (L min$^{-1}$) and $C_i$ (μg m$^{-3}$) are the flow rate of the purged clear air into the Teflon bag
and the mass concentration of VOC$_i$, respectively; M (g) is the dry mass of the enclosed
leaves. The EF$_i$ represents the emission rates of VOC$_i$ under the observed temperature and
PAR.

## 3. Establishment of localized emission rate library

### 3.1. Collection of basal observed emission rates

Our field measurements and the published domestic measurements on plant
species-specific BVOC emission rates in China were integrated to establish the localized
emission rate library. Keywords including "plant volatile organic compounds", "plant VOC
emissions", "BVOCs", "isoprene", and "biogenic VOCs" were utilized to query databases
such as the China National Knowledge Infrastructure (CNKI), Web of Science, Elsevier
ScienceDirect, and Google Scholar. A total of 43 articles on BVOC emission measurements in
China were identified.
All the collected basal data observed under different environment conditions were
normalized to standard condition (Temperature = 30 °C, PAR = 1000 μmol m$^{-2}$ s$^{-1}$) using the
algorithm described in Guenther et al. (1993). Additionally, all emission rates were uniformly
converted to the values by unit of μg g$^{-1}$ h$^{-1}$ (Zhang et al., 2024). In total, we obtained the raw
emission data of 599 plant species. Specifically, the sample size was 845 for isoprene, with
emission rate ranging from 0.002 to 3699.61 μg g$^{-1}$ h$^{-1}$; 846 for monoterpenes with emission
rate ranging from 0.006 to 4281.03 μg g$^{-1}$ h$^{-1}$; 140 for sesquiterpenes, with emission rate
ranging from 0.002 to 143.84 μg g$^{-1}$ h$^{-1}$.
The collected emission rates included the results measured using static enclosure
technique and dynamic one. For static enclosure technique, the branches or leaves were sealed
within an enclosed space for collecting BVOCs. During the enclosure, there is no air in and
out (Préndez et al., 2013; Tsui et al., 2009) , the environment inside the chamber may change



a lot due to the exposure to sunlight and physiological processes of plants, including
temperature, humidity, and carbon dioxide (Stringari et al., 2023, 2024). The changes would
lead to the abnormal BVOC emissions by the enclosed plants. Dynamic enclosure technique
involves the air exchange in the chamber to maintain the environment within close to the
nature (Li et al., 2019). So that its measurements can be expected to present real emission
more. At the earlier stage, static enclosure technique was usually used in China so we
obtained numbers of its observed results. Totally, there were 473 and 421 values of isoprene
and monoterpene emission rate, respectively, from 348 plant species, observed using static
enclosure technique; there were 372, 425, and 140 values of isoprene, monoterpene, and
sesquiterpene emission rate, respectively, from 330 plant species, observed using dynamic
enclosure technique. 79 plant species had observations using both techniques. Despite of the
large uncertainty of observations using static enclosure technique, they were included in the
establishment of our localized rate library, considering that they have a larger sample size and
can display the emission patterns of plants to a certain extent (Stringari et al., 2023). In the
library, the emission rates determined based on the observations using dynamic and static
techniques were defined using different reliability (R) values of 1 and 2, respectively. R-value
= 1 means the higher reliability of emission rates than R-value = 2.

## 3.2. Determination of plant species-specific emission rates

### 3.2.1. Determination of emission categories

All the available normalized isoprene, monoterpene, and sesquiterpene emission rates
from all the plants were separately analyzed. Also, the values observed by dynamic and static
enclosure techniques were separately analyzed. For each library described above, frequency
distribution statistics were conducted. For the observations by dynamic technique, the
isoprene, monoterpene, and sesquiterpene emission rates fell predominantly within 0–600,
0–600, and 0–200 $\mu g\ g^{-1}\ h^{-1}$, respectively, with a sparse distribution of higher emission rates.
For those by static technique, the isoprene and monoterpene emission rates fell predominantly
within 0–300 and 0–50 $\mu g\ g^{-1}\ h^{-1}$, respectively. Firstly, in Figure 1, we divided the emission





range (the x-axis) into various groups, which were further divided into 20 equivalent intervals
separately. Then, we counted the frequencies of values in each interval. The plant emission
rates were inconsistent, but regular in distribution, falling into different intensity levels.
Secondly, ten categories (I–X) were defined for emission rates of isoprene and monoterpenes
measured by static enclosure technique, eleven (I–XI) for monoterpenes measured by
dynamic one, and eight (I–VIII) for sesquiterpenes. Different categories mean different
emission intensities, categories I–XI have emission intensity by low to high level. In our study,
more emission categories were identified than those in previous studies (Klinger et al., 2002;
Wang et al., 2007; Yan et al., 2005).

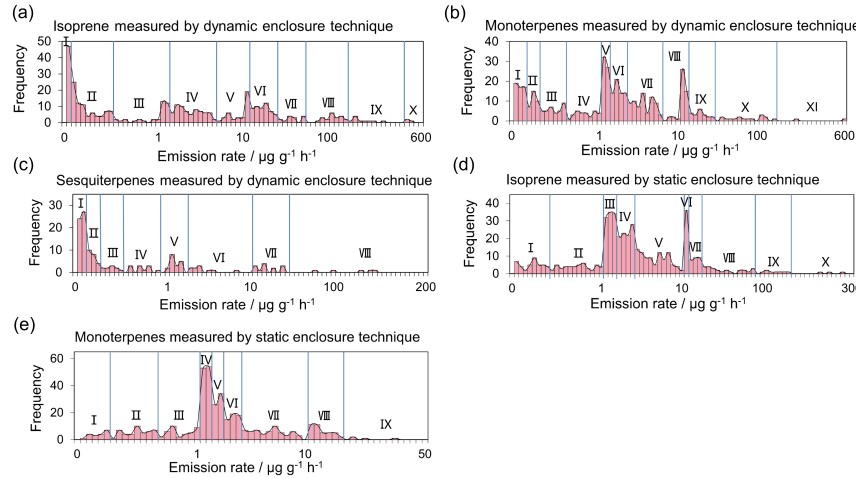


**Figure 1.** Frequency distribution of BVOC emission rates(a–e: Frequency distribution of
BVOC emission rates observed by dynamic (a–c) and static enclosure technique (d–e).)

For each category, the ranges, frequency, mean, and standard deviation (SD) of emission
rates are listed in Table S3. The frequencies of emission rates varied among emission
categories. For the measurements by dynamic enclosure technique, isoprene emission rates
were concentrated in category II, ranging between 0.05–0.5 μg g$^{-1}$ h$^{-1}$, with a frequency of
22%; category VII–X with higher emission intensity totally comprised only 13% of the total
measurements. It indicates that the category with low and moderate emission intensity





included the most plant species and samples. The distribution of monoterpene emission rates
measured by dynamic technique was relatively uniform, with frequencies ranging from 11%
to 16% in most categories. For sesquiterpenes, category I with the lowest emission intensity
had the most measurements, accounting for 36% of the total, indicating the actually lower
sesquiterpene emissions for most plants. Among the emission rates measured by static
technique, the highest frequencies of isoprene and monoterpene emission rates were found in
category III and IV, respectively; Their lowest frequencies occurred in categories with the
highest emission intensity, comprising less than 2% of the total measurements.
The emission rates exhibited a discrete distribution within each emission category,
characterized by large SDs relative to the mean. If the means were considered as the
representative emission rate for each category, large uncertainty would be introduced into the
estimation of emission rate for individual plant species. Therefore, additional statistical
analysis was undertaken to determine the representative emission rates, separately using all
the values in each category which each had a normal distribution. Firstly, 95% confidence
interval (CI) of each emission category was determined through $t$-test (Rivas-Ruiz et al.,
2013). It allowed 95% probability of the actual emission rates locating in each category.
Secondly, the values within the 95% CI for each category were averaged as its representative
emission rate. Thirdly, the emission rate interval for each intensity category was determined
by the ±50% of its representative value. Notably, for the category with the highest emission
intensity, the lower limit of the interval was taken as the representative value due to its limited
samples and high dispersion. Thus, the emission rate intervals and representative values for
each intensity category were obtained specially for each BVOC component and
measurements by dynamic and static techniques separately, which is listed in Table S4.
For the emission categories with lower emission intensity, the representative emission
rates from observations using static enclosure techniques were higher than those from
observations using dynamic technique. While it is opposite for the emission categories with
higher emission intensity. Specially for isoprene, ten emission categories were classified for



observations by both techniques, the representative emission rates from static technique were
higher than those from dynamic one in categories I–V that had lower emission intensity, while
they were lower in categories VI–X that had higher emission intensity.
**3.2.2. Determination of emission rates**
Based on the established detailed categories of emission intensity with more accurate
representative emission rates and intervals, the plant species-specific emission rates were
determined. For a certain plant species, the assignment rule of emission rate is shown as
Figure 2. The assignment is separate for the measurements by dynamic and static techniques.
Then, a localized library including BVOC emission rates for 599 plant species was
constructed, including those estimated based on the measurements by both dynamic and static
enclosure techniques, labeled with R-value of 1 and 2, respectively. This library can be
accessed at https://doi.org/10.5281/zenodo.14557394 (Han et al., 2024).

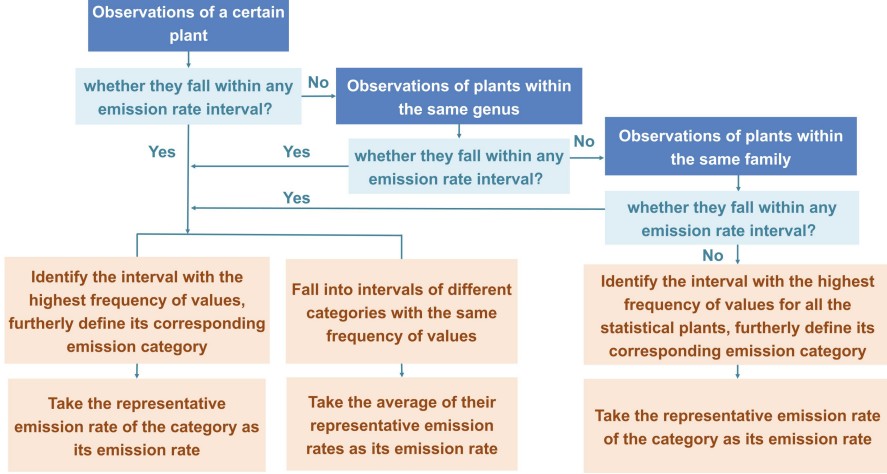


**Figure 2.** Assignment rule of emission rates for a certain plant species.

**3.3. Characteristics of localized emission rate library**
**3.3.1. Emission intensities of plants**
To characterize the emission capacities of different vegetation types, the number of plant





species in each emission category were counted, taking the results with R-value = 1 for
example (Figure S3). The plants were furtherly divided into nine types: evergreen broadleaf
trees, deciduous broadleaf trees, evergreen coniferous trees, deciduous coniferous trees,
evergreen broadleaf shrubs, deciduous broadleaf shrubs, evergreen coniferous shrubs, crops
and herbs. For the isoprene emission, 57% of the total plant species had the low and moderate
intensity by being in categories IV, V and VI. The plants were mainly (69%) evergreen
broadleaf trees, deciduous broadleaf trees, and evergreen broadleaf shrubs. Crops performed
uniform distribution across emission categories II to VI, with low and moderate intensity,
while herbs evenly distributed across emission categories I to X except category II. For the
monoterpene emission, 53% of the total plant species (mainly evergreen broadleaf trees,
deciduous broadleaf trees, evergreen broadleaf shrubs, and deciduous broadleaf shrubs) fell
into categories V, VI, and VII, which could be defined as moderate intensity. Comparatively,
fewer plant species (7% of the total) had higher emission intensities (locating in categories IX,
X and XI). Herb species mostly (67%) had moderate and high monoterpene emission
intensities (locating in categories IV, VII, and VIII), and crop species mostly (85%) had low
and moderate emission intensities (locating in categories III to VII). For the sesquiterpene
emission, 52% of the total plant species had a low intensity by being in categories I and II.
The plants were mainly (78%) evergreen broadleaf trees, evergreen coniferous trees,
deciduous broadleaf trees, and crops. Evergreen broadleaf shrub species mostly (56%) had a
low sesquiterpene emission intensity (locating in category II), emission rates of crop species
mostly (69%) located in categories II and V, with low emission intensities. Deciduous
broadleaf trees and deciduous broadleaf shrubs performed uniform distribution across
emission categories I to VIII.

Generally, most plant species performed low and moderate intensity of isoprene.

Specially, broadleaf plants mostly had a moderate emission intensity, while coniferous plants
mostly had a lower intensity. For the monoterpene emission, both broadleaf and coniferous
plants mostly had a moderate emission intensity. For herbs, the emission intensity of isoprene
and monoterpenes varied greatly and covered low, moderate, and high levels. While the



emission intensity of sesquiterpenes was relatively lower for most plant species, particularly
trees and crops.

**3.3.2. Emission differences among vegetation types**

The distribution of emission rates across various vegetation types is illustrated in Figure

3. Notably, considerable variation existed among vegetation types, characterized by a discrete
distribution. For isoprene, the emission rates of trees were typically concentrated at 0.02–28.5
$\mu g\ g^{-1}\ h^{-1}$, those of shrubs concentrated around 4.2 $\mu g\ g^{-1}\ h^{-1}$, those of crops mainly in
0.20–28.5 $\mu g\ g^{-1}\ h^{-1}$; while emission rates of herbs showed a discrete distribution, with an
average of 26.5 $\mu g\ g^{-1}\ h^{-1}$. Overall, herbs showed the highest isoprene emission, followed by
trees and shrubs, by comparing their means and medians. For monoterpenes, emission rates of
trees and shrubs were primarily concentrated at 1.5–5.8 $\mu g\ g^{-1}\ h^{-1}$, those of crops were mainly
0.46–5.8 $\mu g\ g^{-1}\ h^{-1}$, while those of herbs were evenly distributed, with an average of 17.7 $\mu g$
$g^{-1}\ h^{-1}$. Generally, herbs had the highest monoterpene emission, followed by shrubs, while the
emission of trees and crops were comparatively lower. As for sesquiterpenes, the emission
rates for trees were mainly concentrated at 0.05–0.17 $\mu g\ g^{-1}\ h^{-1}$, secondly in 0.36–4.3 $\mu g\ g^{-1}$
$h^{-1}$, those of shrubs mainly distributed around 0.17 and 1.5 $\mu g\ g^{-1}\ h^{-1}$, those of crops and herbs
were mainly 0.05–0.17 $\mu g\ g^{-1}\ h^{-1}$. Comparatively, trees and shrubs showed the highest
sesquiterpene emission, followed by herbs, crops had the lowest emission. As to the subtypes,
broadleaf plants had relatively higher isoprene emission levels, while coniferous plants had
higher monoterpene emission levels. This may be attributed to the broad and thick leaves of
broadleaf plants, which possess stronger photosynthetic efficiency to produce isoprene
(Benjamin et al., 1996; Li et al., 2021). While the thicker cuticle of coniferous plants can
create favorable conditions for the storage of monoterpenes (Aydin et al., 2014), which are
primarily regulated by temperature and less influenced by light (Tani et al., 2024; Yang et al.,
2021). Moreover, the vegetation types with high sesquiterpene emissions were similar to
those with high monoterpene emissions, which can be explained by the significant correlation
between the emissions of monoterpenes and sesquiterpenes from plants ($P < 0.05$) reported by
Ormeño et al. (2010).

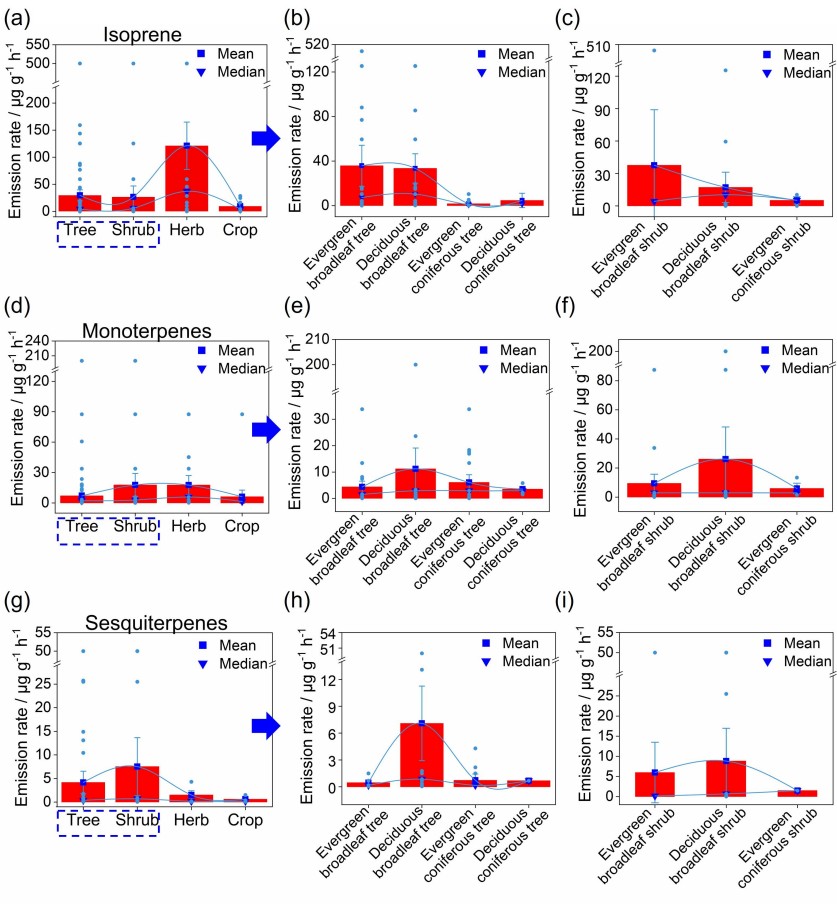

**Figure 3.** Statistics of BVOC emission rates in various vegetation types. (a–i: The distribution
of emission rates for isoprene (a), monoterpenes (d), and sesquiterpenes (g) across vegetation
types of tree, shrub, herb, and crop. The differences in BVOC emission rates between various
subtypes of trees(b, e, h) and shrubs(c f, i). The boxplots display median and mean of the
distribution, the ends of the boxes represent the 25th and 75th percentiles, and outliers are
also displayed in the figure.).

**3.3.3. Interspecific differences in the same family/genus**

Plants within the same family or genus usually share similar morphological and



biological traits (Lun et al., 2020; Wu, 2021). However, BVOC emissions are not only
influenced by genes but the interactions with environment (Benjamin and Winer, 1998),
leading to variations in the components and quantities of BVOC emissions. From our
developed emission rate library, higher BVOC emissions were discovered in the families
Poaceae and Fabaceae, which respectively had higher isoprene and monoterpene emission.
Exceptionally, the isoprene emission rates of crops in Fabaceae were overall higher than those
in Poaceae (Figure 4). Specifically, the emission rate of *Arachis hypogaea* and *Glycine max*
(28.5 µg g$^{-1}$ h$^{-1}$) belonging to Fabaceae was higher than that of *Zea mays* and *Sorghum bicolor*
(16.4 µg g$^{-1}$ h$^{-1}$) belonging to Poaceae. Differences may exist among genera within the same
family. Plants of Poaceae are widely distributed in China even the world (Sun et al., 2024;
Wanasinghe et al., 2024), including the dominant crops like *Triticum aestivum*, *Oryza sativa*,
and *Zea mays*, as well as herbs and bamboo. The plant species, genera, and BVOC emission
rates within Poaceae are listed in Table S5. The evergreen broadleaf trees, the bamboo species,
such as *Fargesia spathacea* and *Bambusa textilis*, are widely distributed and commonly used
for afforestation (Yan et al., 2024). They possessed the highest isoprene emission rate of 500.0
µg g$^{-1}$ h$^{-1}$. In the selection of plant species for the future afforestation, lower-emission bamboo
species like *Bambusa ventricosa* and *Bambusa vulgaris var. striata* should be preferred. Herbs
usually performed higher isoprene and monoterpene emissions, but *Phragmites australis* had
higher sesquiterpene emission. Crops performed higher sesquiterpene emission than other
vegetation types. Also, it is worth mentioning that considerable differences in emission rates
were exhibited even among plants belonging to the same genus. So it may introduce
uncertainties to our developed emission rate library when assigning based on the observations
of all the plants within the same genus or family. Meanwhile, the limited samples in the same
family likely resulted in uncomprehensive conclusion. Anyway, to have more precise
emission rate library, it is necessary to conduct more emission observations in the future to
cover as many plant species as possible. Also, the accuracy of the emission rates in the
developed library derived by this assignment could be verified through field observations in
the near future study.




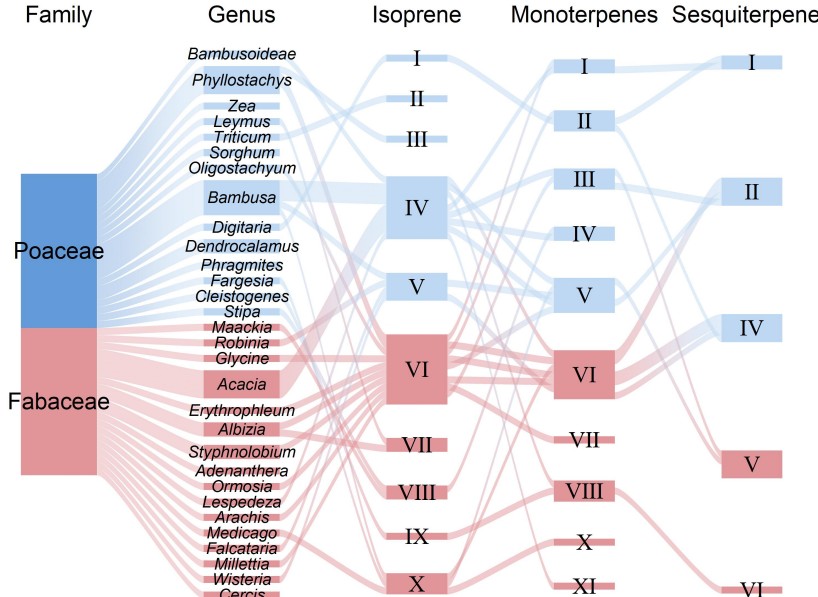


**Figure 4.** Emission categories of the plant species in different genus of families Poaceae and
Fabaceae. (The width of the box represents the number of species it contains, and the colors at
the beginning and end of the connecting lines correspond to the the two ends they connect.)

**3.3.4. Variability in emission rates derived from dynamic and static enclosure**
**measurements**
In the localized emission rate library, the subsets of emission rates with R-value = 1 and
2 were separately established. For isoprene emission rates, 51% of plant species exhibited
higher values with R-value = 2 than those with R-value = 1, among these plants, 66% had
emission rate of 0.02–4.2 µg g⁻¹ h⁻¹ (in lower emission intensity). In contrast, for plants with
emission rates of R-value = 1 higher than those of R-value = 2, 78.1% had emission rates of
28.5–500.0 µg g⁻¹ h⁻¹ (in moderate and high emission intensity). For monoterpene emission
rates, 49% of plant species displayed higher values with R-value = 2 than those with R-value





= 1, with all emission rates below 13.4 µg g$^{-1}$ h$^{-1}$ (in lower emission intensity). In contrast, for
plants with emission rates of R-value = 1 higher than those of R-value = 2, 48% had emission
rates of 13.4–200.0 µg g$^{-1}$ h$^{-1}$ (in moderate and high emission intensity). To be concluded, the
emission for plants with high intensity may be underestimated when measuring by static
enclosure technique, while that for plants with low intensity may be overestimated. The
discrepancy between the emission rates derived from dynamic and static enclosure
measurements is likely attributed to the following: static enclosure technique may induce a
large buildup of BVOCs and release of stressed compounds due to the environment changes
within the chamber, while it has lower detection limit that can make more compounds being
detected (Li et al., 2019), leading to the higher emission rate than dynamic measurement for
the plants with low emission intensity; the plants with high emission intensity also have
strong transpiration, leading to condensation of moisture on the walls within the static
enclosure, then the closure of plant stomata and reduced emission (Kfoury et al., 2017), also
high concentration of BVOCs may undergo reactions and degradations in the chamber
(Antonsen et al., 2020), together contributing to the underestimates by static technique for the
plants with high emission intensity.

## 4. Application of localized emission rate library

### 4.1. BVOC emission simulation

MEGANv3.2 was applied to estimate BVOC emissions. The variables driving

MEGANv3.2 include vegetation data, meteorological parameters, and emission rates.
Specifically, the vegetation data include the distribution of four growth forms, their species
speciation, ecological types, canopy types, and leaf area index (LAI). In the study, the
database of high-resolution vegetation distribution (HRVD) with a horizontal resolution of 1
km × 1 km established by Cao et al. (2024) (https://zenodo.org/records/10830151) was used
to produce the distribution of growth forms and canopy types. It integrates the multiple
sources of land cover data including the China multi-period land use/cover change remote



sensing monitoring data set (CNLUCC) (Xu et al., 2020), MODIS MCD12Q1 land cover
product (Friedl and Sulla-Menashe, 2019), as well as Vegetation Atlas of China (1:1,000,000)
and shows a significant correlation with the filed investigation. The vegetation speciation was
derived from the Vegetation Atlas of China (1:1,000,000). LAI was from the MODIS version
6.1    LAI    product    reprocessed    by    Lin    et    al.    (2023)
(http://globalchange.bnu.edu.cn/research/laiv061) and furtherly updated based on the HRVD.
Hourly meteorological fields driving MEGANv3.2 were simulated by the Weather Research
and Forecasting (WRF) v3.8.1. The simulation covered the whole China with a horizontal
resolution of 36 km × 36 km and was for the year 2020.

The plant species-specific emission rates in the simulation were derived from our

developed localized library. To match the input for MEGANv3.2, where monoterpenes and
sesquiterpenes were categorized into five and two categories, respectively, the emission rates
of monoterpenes and sesquiterpenes were assigned to separate categories based on the
relationships of the global ones in MEGANv3. Totally, emission rates of 283 plant species
were updated, including 257, 280, and 101 species for isoprene, monoterpenes, and
sesquiterpenes, respectively. Specifically, 202 plant species had the emission rates with
R-value = 1, while the additional 81 species had those with R-value = 2.

Four simulations were established to explore the impacts of quality of emission rates on

BVOC emissions, as shown in Table 1. All the simulations had the same input other than
emission rates. In Simulation 1, our developed emission rate library was fully applied
including the emission rates with R-value of both 1 and 2, but those with R-value = 1 were
selected preferentially, then supplemented by those with R-value = 2. This simulation made
the emission rates to be localized as much as possible. Due to the higher accuracy of emission
rates with R-value = 1 than those with R-value = 2, in Simulation 2, only those with R-value
= 1 were applied, which would be expected to have more precise estimates. To investigate the
impact of emission rates in different quality on the BVOC emission estimates, the results of
Simulation 1 and 3, which was set by applying the emission rates with R-value = 2
preferentially, then supplemented by those with R-value = 1. Simulation 4 utilized the global



library in MEGANv3.2. By comparing results in Simulation 1 and 4, the differences could be
explored after localizing the emission rates.
**Table 1.** Simulation schemes for BVOC emission estimation.

| Simulations | Emission rate | Description |
|---|---|---|
| Simulation 1 | Localized library, emission rates with R-value = 1 preferentially and R-value = 2 supplementally | To include more domestic emission rates |
| Simulation 2 | Localized library, emission rates with R-value = 1 | To include domestic emission rates with higher reliability |
| Simulation 3 | Localized library, emission rates with R-value = 2 preferentially and R-value = 1 supplementally | To explore the impact of emission rates with different reliability on the emission estimates, by comparing with Simulation 1 |
| Simulation 4 | Global library in MEGANv3.2 | To simulate the emissions using global emission rates without localization |


## 4.2. BVOC emissions in China

Based on the results in Simulation 1 where our developed emission rate library was fully
applied including the emission rates with R-value of both 1 and 2 (as in Table 1), the annual
total BVOC emission in China for the year 2020 was 27.70 Tg. Its composition is shown in
Figure S4. In the four categories of BVOCs, other VOCs contributed the most, accounting for
47% of the total emissions. The large contribution was attributed to their large numbers of
compound species, comprising more than half of the total simulated compounds in
MEGANv3.2. Isoprene and monoterpenes exhibited comparable contributions, accounting for
23% and 25% of the total, respectively. Specifically, isoprene, butane, and isobutene emerged
as the most substantial contributors to BVOC emissions, jointly accounting for 44%.
BVOC emissions in China exhibited large spatial variations, higher in the southeast and
lower in the northwest (Figure 5). Specifically, the high emissions in the Southeast Hill,




Yunnan-Guizhou Plateau, and Taiwan Province locating in the southeastern China were
primarily attributed to the extensive coverage of evergreen broadleaf trees (Cai et al., 2024).
Among them, the widely distributed plants *Quercus fabri*, *Bambusa textilis*, and *Lithocarpus*
*amygdalifolius* had higher isoprene emission rates of 85.5, 500.0, and 125.4 μg g⁻¹ h⁻¹,
respectively. Suitable environments characterized by high temperatures were also major
contributors to the high emissions in these regions (Duan et al., 2023). Furthermore, the
Greater and Lesser Khingan Mountains and Changbai Mountains were rich in forest resources,
including both coniferous and broadleaf trees, accounting for over 77% of the total vegetation
distribution in those regions, resulting in relatively higher BVOC emissions. The North China
Plain and Sichuan Basin, with their widespread crop cultivation, accounting for 74% and 54%
of the total vegetation coverage, also exhibited high emissions. The lower emissions in the
northwest were likely due to the predominance of herb species with lower emission rates,
such as *Festuca ovina*, *Krascheninnikovia compacta*, and *Elymus nutans*.

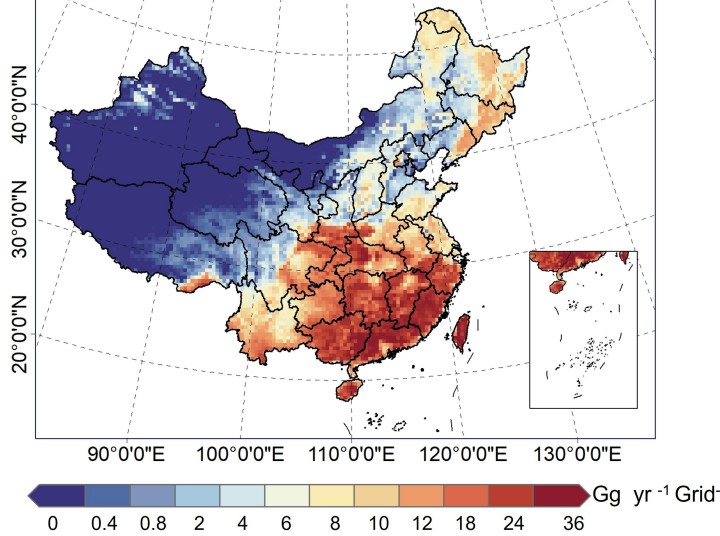

**Figure 5.** Spatial distribution of BVOC emissions in China in 2020 (Spatial distribution of
BVOC emissions estimated based on the localized emission rate library in China in 2020.)
Compared to the results in Simulation 4, BVOC emission was 18% higher after updating





the emission rates applying our developed library than that (23.44 Tg) applying the global
emission rates without localization. By BVOC categories, the emissions of isoprene,
monoterpenes, and sesquiterpenes increased by 55%, 29%, and 48%, regspectively. The
contribution of isoprene, monoterpenes, sesquiterpenes, and other VOCs to total BVOC
emission changed from 18%, 23%, 4%, and 55% to 23%, 25%, 5%, and 47%, respectively.
Discrepancy was also found in the spatial distribution (Figure 6c). In the southeastern China,
especially the Sichuan Basin, Simulation 1 showed higher emission than Simulation 4. It was
likely due to the widespread distribution of crops, which had higher emission rates in our
library compared to the global one. Conversely, in the western and northeastern China,
particularly in the Greater and Lesser Khingan Mountains and Changbai Mountains,
emissions in Simulation 1 were less than Simulation 4. It was mainly because of the extensive
distribution of the genera *Pinus* and *Betula*. Their isoprene emission rates with R-value = 1
were used in Simulation 1 which were 80% and 86% lower than the global ones, respectively.
Also, the plants of the two genera were widespread, accounting for 70% of the total
vegetation coverage area in the Greater and Lesser Khingan Mountains and Changbai
Mountains. To verify the performance in simulating the spatial distribution of emissions
before and after applying the localized and global emission rates, the correlation between
emission and observed formaldehyde (HCHO) vertical column density (VCD), was analyzed.
HCHO in the atmosphere can serve as a reliable proxy for tracing the biogenic source of
isoprene, especially in the summer (Liu et al., 2024). Here, the Sentinel-5p TROPOspheric
Monitoring Instrument (TROPOMI) Spaceborne HCHO products that can be accessed
through the Google Earth Engine (GEE) platform (https://code.earthengine.google.com/) were
used. The isoprene emissions in July in Simulation 1 correlated stronger with HCHO
concentration spatially (correlation coefficient = 0.73, P < 0.05) than Simulation 4
(correlation coefficient = 0.67, P < 0.05). It suggests that the application of our localized
emission rate library could simulate the spatial variations of BVOC emissions better. Using
the global emission rate library, there might be an underestimate in the south and overestimate
in the northeast and west, which could be abated by updating the localized emission rates.





### 4.3. Impact of emission rates with different reliability on BVOC emission estimates

To apply the more accurate emission rates, Simulation 2 was conducted employing only the emission rates with R-value = 1 from the localized library. As to the plant species having emission rates with R-value = 2 in Simulation 1, the global emission rates were assigned in Simulation 2. Compared to the estimation from Simulation 1, there was a similar emission (27.46 Tg). By BVOC categories, the isoprene emission increased by 4%, while monoterpene emission decreased by 2%; the emissions of sesquiterpenes and other VOCs remained unchanged. The BVOC composition changed little. Spatially, in most regions of China, the emissions of Simulation 1 were a little lower than those from Simulation 2 by -1−0 Gg yr$^{-1}$ grid$^{-1}$ (Figure 6a). The main reason for this discrepancy was primarily attributed to the fact that the emission rates for plants updated by those with R-value = 2 in Simulation 1 were generally lower than the global ones in Simulation 2. Among these plants, herbs accounted for 84%, while trees and shrubs only accounted for 3% and 14%, respectively. The average isoprene and monoterpene emission rates of these herb species derived from our library were 7% and 67% lower than those from the global library, respectively. In contrast, the emissions of Simulation 2 exceeded those of Simulation 1 in certain areas by 0−6 Gg yr$^{-1}$ grid$^{-1}$, where were concentrated in the South China, Lesser Khingan Mountains, and Changbai Mountains. This was primarily due to the distribution of herbs belonging to the *Carex* genus, whose isoprene emission rate with R-value = 2 were 24% higher than the global values. These plants comprised 31% of the total herb coverage. The above made the national total BVOC emissions change a little when excluding the emission rates with lower reliability in the emission estimates.

Furthermore, to investigate the impact of using emission rates with R-value = 1 versus R-value = 2 on the estimated emission, Simulation 3 was conducted by using those with R-value = 2 preferentially and then those with R-value = 1 supplementally. The results in Simulations 1 and 3 were compared. Simulation 3 gave an increased BVOC emissions by 7%.





By BVOC categories, the emissions of isoprene and monoterpenes rose by 17% and 11%,
respectively, those of sesquiterpenes and other VOCs remained similar. Their contributions to
the total BVOC emissions changed little. Spatially, in most regions, the emissions in
Simulation 3 were higher than those in Simulation 1 (Figure 6b), particularly, in the Sichuan
Basin. There, *Oryza sativa*, one crop species, accounted for 93% of the total crop coverage.
Its isoprene and monoterpene emission rates with R-value = 2 were 1.2 and 14.2 μg g$^{-1}$ h$^{-1}$,
much higher than those with R-value = 1 (0.18 and 5.8 μg g$^{-1}$ h$^{-1}$). In the Lesser Khingan
Mountains and Changbai Mountains, the isoprene emission rate for the widely distributed
genus *Larix* with R-value = 2 was 166% higher that with R-value = 1; for the species *Pinus*
*koraiensis*, the monoterpene emission rate with R-value = 2 was 390% higher. Conversely, in
the areas where herbs were widely distributed, especially in the northwest of China, the
emissions simulated in Simulation 3 were lower than those in Simulation 1. It was likely due
to the fact that for most herb species, the emission rates with R-value = 2 were lower than
those with R-value = 1. For instance, the applied isoprene emission rates for the genera *Stipa*,
*Cleistogenes*, and *Leymus* in Simulation 1 were 125.8, 258.9, and 59.5 μg g$^{-1}$ h$^{-1}$, respectively,
while they were 1.2, 1.2, and 4.2 μg g$^{-1}$ h$^{-1}$, respectively, in Simulation 3. For the estimated
emission in Simulation 3, the correlation between it and observed HCHO VCD was analyzed.
Their correlation coefficient was 0.63 (P < 0.05). Meanwhile, the correlation coefficient (0.72)
for the emission estimated in Simulation 2 was also higher than that in Simulation 3. So, it
can be concluded that the accuracy of the estimation decreased after introducing the emission
rates with R-value = 2. Notably, compared to Simulation 2, Simulation 1 also introduced the
measurements with R-value = 2, however, most of the species (84%) were herbs, and the
coverage of them were limited. The above made the similar national total BVOC emissions
and spatial accuracy of the two simulations. Overall, the usage of emission rates with R-value
= 2 could overestimate the total BVOC emissions in China. Therefore, in future, more
emission observations with high reliability using dynamic technique are strongly encouraged
to furtherly improve the accuracy of the localized emission rate library and emission
inventory.

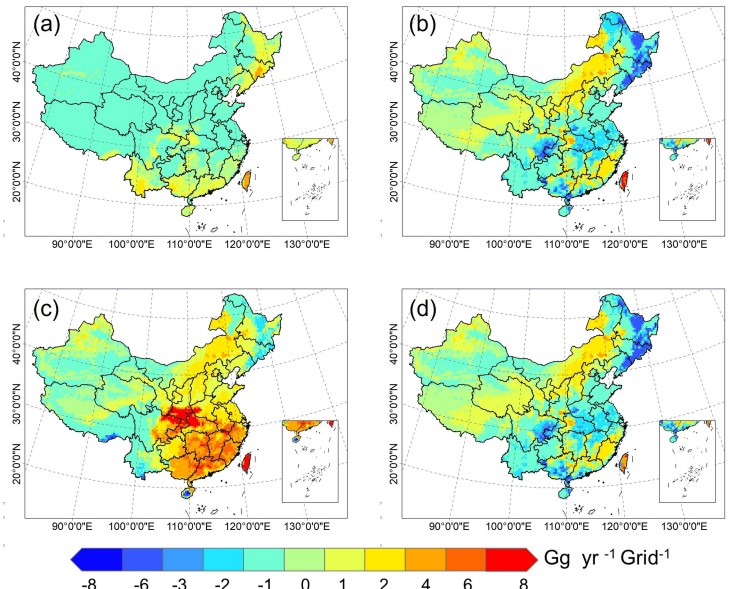

**Figure 6.** Spatial distribution of differences among BVOC emissions simulated using different emission rates. ((a–d): Simulation 1 minus Simulation 2 (a); Simulation 1 minus Simulation 3 (b); Simulation 1 minus Simulation 4 (c); Simulation 2 minus Simulation 3 (d).).

## 5. Conclusion

By integrating our and reported local field measurements, a statistical approach of classifying emission categories and determining the plant species-specific emission rates that can be used in BVOC emission inventory compilation was developed. It worked out more detailed categories of emission intensity, accurate emission rate intervals and representative values compared to previous studies, namely ten, ten or eleven, and eight categories respectively for isoprene, monoterpene, and sesquiterpene emission rates. The detailed categories for emission intensity can furtherly improve the determined representative emission rates. Based on this, a localized plant species-specific BVOC emission rate library for China, including isoprene, monoterpene, and sesquiterpene emission rates for 599 plant species. In this library, observations by both dynamic and static techniques were included and



separated with different reliability. Variability was found in the emission rates deriving from
dynamic and static enclosure measurements. Specifically, the measurements by static
enclosure technique may underestimate the emissions of plants with higher emission intensity
and overestimate the emissions of plants with lower emission intensity. Analyzing the
emission rates deriving from the dynamic technique measurements in our library, by
comparing the means and medians for different vegetation types, herbs showed the highest
isoprene emission level, followed by trees and shrubs; herbs also had the highest monoterpene
emission level, followed by shrubs, while trees and crops were comparatively lower; trees and
shrubs showed the highest sesquiterpene emission level, followed by herbs and crops.
Interspecific differences were exhibited in the same type/family/genus.

Furtherly, our localized emission rate library was applied in the China's BVOC emission

inventory compilation and the performance was evaluated. By updating the localized
emission rates, the simulated BVOC emission in China in 2020 was 27.70 Tg, 18% higher
than that using the global emission rate library. Isoprene, monoterpenes, sesquiterpenes, and
other VOCs contributed 23%, 25%, 5%, and 47% to the total emission, respectively. It had a
better performance in emission estimation with the higher correlation coefficient of 0.73 ($P <$
0.05) between isoprene emission and HCHO VCD observations spatially. The underestimates
in the south and overestimates in the northeast and west when using global emission rates
could be abated by updating the localized ones. Using emission rates with different reliability
could result in different emission estimates and model performance. The usage of emission
rates measured by static enclosure technique could decrease the accuracy of estimation and
result in an overestimation to the BVOC emission. Therefore, in BVOC emission inventory
compilation, it is suggested to use emission rates measured by dynamic enclosure technique
more to achieve more accurate results.

Although our developed localized emission rate dataset is benefit for improving the

accuracy of emission inventory, uncertainties still exist in the dataset and its application.
Firstly, the dataset still includes limited plant species so that it is hard to cover all the plants in
China. Researchers have to use the global emission rates for the plants without localized



observation. Secondly, it may introduce errors when assigning emission rates based on the
observations of plants within the same genus or family. For the emission rates with R-value =
1, 16% were allocated by genus and 13% by family; for those with R-value = 2, 3% were
allocated by genus and 5% by family. Thirdly, for monoterpene and sesquiterpene emission
rates separately, some of the raw observed results are the sum of their studied dominated
compounds rather than the whole category of monoterpene and sesquiterpene. Therefore, in
the application of our dataset, there may be an underestimation for their emissions.
Meanwhile, the MEGAN model requires more detailed category for them, it is better to
conduct compound-specific observation and obtain their emission rates. Fourthly, the
determined emission category can be more detailed and the emission rate intervals and
representative values can be more accurate if we have more local observation samples. The
above uncertainties would be reduced by including more reliable local emission
measurements specifically by plant species and compounds in the future. Notably, although
the current uncertainties, our study starts the effort to establish a reliable localized dataset of
BVOC emission rates used in inventory compilation. Undeniably it helps to improve the
accuracy of the determined emission rates and furtherly the emission estimates. Meanwhile,
our developed statistical approach can be extended to the establishment of localized BVOC
emission rate dataset for other regions.

## Data availability

All datasets used in this study are publicly available. The localized plant species-specific

BVOC    emission    rate    dataset    is    available    from    *Zenodo*    at
https://doi.org/10.5281/zenodo.14557394. Spaceborne HCHO products is available from
Google Earth Engine platform at https://code.earthengine.google.com. Database of
high-resolution    vegetation    distribution    (HRVD)    is    available    from    *Zenodo*    at
https://zenodo.org/records/10830151. LAI is available from the MODIS version 6.1 LAI
product at http://globalchange.bnu.edu.cn/research/laiv061.




## Author contributions

LL conceived and designed the study, HH performed the data analysis, carried out the
model simulations, and drafted the manuscript with the help of RS, CN, and YK. YJ
conducted data field measurements and collection, YW provided analysis of the measured
data, All authors reviewed and commented on the paper.

## Acknowledgments

This study was supported by the National Key Research and Development Program of
China (2023YFC3710200), Development Plan for Youth Innovation Team of Colleges and
Universities of Shandong Province (2022KJ147), and National Natural Science Foundation of
China (42075103).

## Financial support

This study has been supported by the National Key Research and Development Program
of China (2023YFC3710200), Development Plan for Youth Innovation Team of Colleges and
Universities of Shandong Province (2022KJ147), and National Natural Science Foundation of
China (42075103).

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
