# Peer review of "A localized, plant-species-specific BVOC emission rate library for China using a statistical analysis of field measurements"

_EGUsphere, 2025_

## Author Comment (AC1)

**Response to Referee #1:**

The study by Han et al. presents the development of a plant species-specific BVOC emission rate library for China, based on a combination of literature data and new measurements. The authors employ a statistical approach to classify emission intensities, define representative emission rates, and assign reliability levels depending on the measurement technique. They then implement the new emissions in MEGAN v3.2 and assess its performance against formaldehyde satellite observations. The topic is timely and relevant, as accurate biogenic emission inventories remain a major uncertainty in atmospheric chemistry. The manuscript compiles an impressive amount of local data that will be valuable for the community. However, in its current form, the study is not suitable for publication, and substantial revisions are required before it can be considered further.

**Response:**

Thank you for your positive and constructive comments. In this revision, we have strived to achieve the greatest improvement and hope you agree with our modification.

**General comments**

1. The most urgent and critical issue is the language. The manuscript is written in a way that makes it very difficult to follow, to the point that the scientific message is often obscured. The text frequently becomes repetitive, and in many cases, it is challenging to understand what the authors intend to communicate. The entire manuscript must be carefully revised to improve clarity, flow, and scientific precision, ideally with the assistance of a fluent English speaker or a language model. Enhancing the clarity of the text would significantly increase the scientific value and accessibility of the study.

**Response:**

Thank you for your valuable comment. We are sorry for the poor writing in the manuscript. In revised manuscript, we check through the article and rewrite the bad sentence. Besides, English is edited by Elsevier Language Editing Services. The certificate is as follows.

[Figure]

**Certificate of Elsevier**
**Language Editing Services**

**The following article was edited by Elsevier Language Editing Services:**

**A localized plant species-specific BVOC emission rate library
of China established using a developed statistical approach
based on field measurements**

**Ordered by:**

**Lingyu Li**

**Estimated Delivery date:**

**2025-12-04**

**Order reference:**

**ASLESTD1120118**

[Figure]

[Figure]

The changes caused by the revision of English are not marked in revised manuscript, and are not listed here due to the limited space.

2. Despite the breadth of the compiled data, the scientific depth remains limited. The classification of emissions into categories and the assignment of representative rates read more as a statistical simplification than a mechanistic interpretation of BVOC variability. As a result, this otherwise valuable database appears as a fragile inventory, with its reliability tightly dependent on methodological choices such as normalization procedures and the inclusion of static chamber data.

**Response:**

Thank you for your insightful comment.

(1) In previous studies, the process of determining emission categories, representative emission rates, and ranges is not straightforward and lacks theoretical evidence. So, in our study, we adopt the statistics to do the categorization and determine the representative emission rates. That can be expected to improve the accuracy of the assigned emission rates for plants. It is actually the statistical results, but the categorization can show the BVOC

emission variability to a certain extent: BVOC emissions present as different intensity, for example very low, low, moderate, high, very high, etc. The emission intensity usually differs among plant species. For instance, from our statistical results, a majority (57%) of plant species exhibited low-to-moderate isoprene emission intensity (Categories IV–VI). For monoterpenes, 53% of species were primarily characterized by moderate intensity (Categories V–VII). As for sesquiterpenes, over half (52%) of species demonstrated low emission intensities (Categories I–II).

The above has been illustrated in the manuscript: "First, the process of determining emission categories, representative emission rates, and ranges is not straightforward and lacks theoretical justification." in line 67-69; "First, most plant species (57%) exhibited low-to-moderate isoprene emission intensities (Categories IV–VI)." in line 277-278; "Besides, monoterpene emissions were primarily characterized by moderate intensity (Categories V–VII), encompassing 53% of all species." in line 282-283; and "As for sesquiterpene emissions, over half of the plant species (52%) demonstrated low sesquiterpene emission intensities (Categories I–II)," in line 288-290. For explanation, in line 201-204 of revised manuscript, "The plant emission rates were inconsistent, but regular in distribution, falling into different intensity levels." is changed to "Although individual plant emission rates were inconsistent, they exhibited a clear regularity in distribution, forming distinct intensity levels. Most measurements clustered around a mean value (the peak of the curve), revealing an underlying statistical structure despite individual variability.".

(2) As you commented, the results and reliability of our library tightly depend on the normalization procedures and the raw data. Firstly, the coarser classification may result in overestimation or underestimation of emission rates. By contrast, more detailed the category, more real the emission rate. In previous studies, it was usually 5–7 categories, while ours are more detailed by 10–11 categories based on quantities of data sample and statistical analysis. Secondly, although the static measurements have larger uncertainty, they provide a larger sample size of local emission rates which makes them still be able to give us valuable insights into emission patterns of local plant species. For example, in our library, for trees and crops,

emission rates with R-value = 1 cover 93% of the total tree area and 94% of the total crop area in China; for shrubs and herbs, the coverage of plants with emission rate of R-value = 1 account for 34% and 21% of their respective coverage areas. Static data are more abundant for shrubs and herbs. So, to maximize the localization, the static data can be included in our library. Their inclusion can also allow us to understand the BVOC emission features and quantify the variability between both measurement methods.

The above has been illustrated in the manuscript: "Third, most studies used coarse classifications of emission, typically five to seven categories (Klinger et al., 2002; Yan et al., 2005), which may result in imprecise classifications and overestimation or underestimation of emission rates for specific plant species." in line 73-76; "Second, ten categories (I–X) were defined for emission rates of isoprene and monoterpenes measured by the static enclosure technique, eleven (I–XI) for monoterpenes measured by dynamic one, and eight (I–VIII) for sesquiterpenes. Different categories represent different emission intensities, with categories I–XI representing emission levels from low to high. In the present study, more emission categories were identified than those in previous studies (Klinger et al., 2002; Wang et al., 2007; Yan et al., 2005)." in line 204-210; "Despite the large uncertainties associated with the static enclosure technique, these observations were included in establishing our localized rate library because of their larger sample size and ability to display plant emission patterns to a certain extent (Stringari et al., 2023)." in line 183-186. In line 444-450 of the revised manuscript, "The application of emission rates with R-value = 1 was assessed by calculating the plant species coverage percentage of the total vegetation. Emission rates with R-value = 1 cover a high percentage of the dominant vegetation, specifically 93% of the total tree area and 94% of the total crop area. In contrast, their coverage is substantially lower for shrubs and herbs, with 34% and 21% of their respective areas. This is a common challenge in regional BVOC modeling, as comprehensive field measurements for all shrub and herb species are often limited." is added.

3. The procedure for incorporating the new library into MEGAN is insufficiently described. Since MEGAN operates on PFTs rather than individual species, it is unclear how

species-level data were aggregated, mapped, or reconciled with the model structure. In addition, the description of the simulations should be thoroughly revised, as it is difficult to follow how the new emission factors were actually tested. Because much of the subsequent discussion and interpretation depend on these simulations and their linkage to the emission classification, the current lack of clarity makes it challenging to evaluate the results. This section needs substantial restructuring and precise methodological explanation.

**Response:**

Thank you for your valuable comment, which allows us to clarify a key aspect of our methodology.

(1) For the mapping of our plant species-level emission factors into the MEGANv3.2 calculation. In earlier versions of MEGAN (such as 2.1) typically operate using PFT-averaged emission factors. In MEGAN version 3.2 that we used, both PFT distribution and detailed vegetation species composition in grids are inputted. Based on the vegetation composition, the gridded PFT-averaged emission factors can be calculated from the species-specific emission factors using the emission factor processing module of MEGANv3.2, and are then included in the emission calculator. Notably, the plant species in our library don't cover all the plants in the vegetation speciation file, for plants being not included in our library, we assign their emission factors by the global ones.

We apologize for the confusion caused by our initial wording. The manuscript has been revised to clearly emphasize this mapping methodology. In line 432-438, "The plant species-specific emission rates in the simulation were derived from our developed localized library." is changed to "In MEGAN version 3.2 used in this study, both PFT distribution and detailed vegetation species composition in grids are entered. Based on the vegetation composition, the gridded PFT-averaged emission factors can be calculated from the species-specific emission factors using the emission factor processing module of MEGANv3.2 and are then included in the emission calculator. Notably, plant species in our library did not cover all the plants in the vegetation speciation file; for species not included in our library, we assigned their emission factors using the global values.".

(2) For the simulation setup. We are sorry for the unclear description of the simulation setups. Following your advice, we have now thoroughly revised the section to clarify the objectives, data sources, and priority rules of emission rates for each simulation experiments in a more logical and structured manner. And we have modified Table 1 to summarize the key design of all simulations to improve clarity. In line 451-469 of revised manuscript, "Four simulations were established to explore the impacts of quality of emission rates on BVOC emissions, as shown in Table 1. All the simulations had the same input other than emission rates. In Simulation 1, our developed emission rate library was fully applied including the emission rates with R-value of both 1 and 2, but those with R-value = 1 were selected preferentially, then supplemented by those with R-value = 2. This simulation made the emission rates to be localized as much as possible. Due to the higher accuracy of emission rates with R-value = 1 than those with R-value = 2, in Simulation 2, only those with R-value = 1 were applied, which would be expected to have more precise estimates. To investigate the impact of emission rates in different quality on the BVOC emission estimates, the results of Simulation 1 and 3, which was set by applying the emission rates with R-value = 2 preferentially, then supplemented by those with R-value = 1. Simulation 4 utilized the global library in MEGANv3.2. By comparing results in Simulation 1 and 4, the differences could be explored after localizing the emission rates." is changed to "To systematically evaluate the performance of the localized emission library developed in this study, four simulation experiments (Simulation 1–4) were conducted under identical model configurations, with the only variation being the source of emission rates. Simulation 1 incorporated all available species-specific emission rates in our localized library, prioritizing those with high accuracy (R-value = 1) and supplementing with lower-accuracy records (R-value = 2) where necessary, thereby maximizing localization. Simulation 2 used only R-value = 1 emission rates to establish a baseline under the most reliable data scenario. Simulation 3 also employed the full localized library but favored R-value = 2 data and supplemented it with R-value = 1. This design enabled a controlled assessment of data quality influence through comparison with Simulation 1. In Simulations 1–4, for plants without assigning localized emission rates,

global data were used. Finally, Simulation 4 relied on the default global emission rate library embedded in MEGANv3.2, serving as a reference to quantify the net effect of emission rate localization when compared with Simulation 1. A summary of the simulation design is provided in Table 2 for clarity.".

**Table 2.** Simulation experiments for evaluation of the localized emission rate library

| Simulations | Emission rate | | | Objective |
| | Local data with R-value =1 | Local data with R-value =2 | Global data in MEGANv3.2 | |
| --- | --- | --- | --- | --- |
| Simulation 1 | √√√ | √√ | √ | To maximize localization and represent the basic estimate using all available local data |
| Simulation 2 | √√√ | - | √√ | To provide an estimate based solely on the high-quality local data |
| Simulation 3 | √√ | √√√ | √ | To investigate the sensitivity of results to the data quality of emission rates, comparing with Simulation 1 |
| Simulation 4 | - | - | √√√ | To serve as a benchmark for quantifying the impact of localization |

Note: The number of √ symbols shows the priority from high to low. Taking Simulation 1 as an example, the plant species-specific emission rates are assigned from the localized library with R-value = 1 primarily (labeled √√√), then those with R-value = 2 as a supplement (labeled √√), and global data are used for plants without localized emission rates (labeled √).

**Specific comments**

1. L18. Aerosol pollution is mainly due to direct emissions. Please revise to clarify that

BVOCs contribute significantly to the production of ozone and secondary organic aerosols.

**Response:**

Thank you for your valuable suggestion. We are sorry for the inaccurate expression. In line 17 of revised manuscript, "ozone and aerosol pollution" is changed to "ozone and secondary organic aerosol pollution".

2. L26. Consider replacing the performance with sensitivity or implications.

**Response:**

Thank you for your valuable suggestion. We are sorry for the inaccurate expression. In line 26 of revised manuscript, "performance of the developed library" is changed to "implications of the developed library".

3. L31-33. Consider moving this sentence to the conclusions or removing it entirely, keeping the final paragraph of the introduction focused on the study objectives.

**Response:**

Thank you for your valuable suggestion. In the revised manuscript, "More local emission observations in higher reliability are encouraged to improve the accuracy of emission rates and further the emission estimates" is removed to keep the final paragraph of the introduction focused on the study objectives.

4. L39. The cited reference is not relevant to BVOC reactivity. Please consider citing a more appropriate chemical reactivity source, such as https://doi.org/10.1021/cr0206420

**Response:**

Thank you for your valuable suggestion. We are sorry for the inaccurate citation. In line 35 of revised manuscript, "(Vella et al., 2023)" is changed to "(Atkinson et al., 2003)".

Reference:

Atkinson, R. and Arey, J.: Atmospheric degradation of volatile organic compounds, Chem. Rev., 103, 4605–4638, https://doi.org/10.1021/cr0206420, 2003.

5. L58 and below. Replace errors with uncertainties. The term error would only be appropriate if the correct reference values were known.

**Response:**

Thank you for your valuable suggestion. We are sorry for the inaccurate expression. In line 54, 58, 640 of revised manuscript, "errors" is changed to "uncertainties".

6. L86-87. A reference is needed.

**Response:**

Thanks for your careful review and valuable suggestion. According to your comment, in line 82 of revised manuscript, the reference (Batista et al., 2019) is added to support this claim.

Reference:

Batista, C. E., Ye, J., Ribeiro, I. O., Guimarães, P. C., Medeiros, A. S. S., Barbosa, R. G., Oliveira, R. L., Duvoisin, S., Jardine, K. J., Gu, D., Guenther, A. B., McKinney, K. A., Martins, L. D., Souza, R. A.F., and Martin, S. T.: Intermediate-scale horizontal isoprene concentrations in the near-canopy forest atmosphere and implications for emission heterogeneity, P. Natl. Acad. Sci. USA, 116, 19318–19323, https://doi.org/10.1073/pnas.1904154116, 2019.

7. L110/Figure S2. I recommend moving this figure to the main text.

**Response:**

Thank you for your suggestion. We have moved Figure S2 to the revised manuscript as Figure 1.

[Figure]

**Figure 1.** Schematic of dynamic enclosure technique. Vacuum pump was used to introduce

air to the system; silicone rubber, potassium iodide, and activated carbon were used to remove $O_3$ and VOCs from the air; a mass flow meter was used to control flow rate; a temperature sensor and light quantum sensor were used to record temperature and photosynthetically active radiation; after equilibrium, the gases in the bag were collected into adsorption tubes using an air-sampling pump.

8. L110-113. Teflon bags are widely used in BVOC research due to their chemical inertness and negligible BVOC emissions. Please clarify how you further "passivated" them.

**Response:**

Thanks for your careful reading and valuable comments. We are sorry that we used a wrong expression by the word "passivated". No additional chemical passivation treatment was applied to the Teflon bags. As you correctly pointed out, Teflon itself exhibits excellent chemical inertness, and extensive literature has demonstrated that its adsorption/desorption of common BVOCs is negligible. Therefore, what we intended to convey in the original text was that the inherent inertness of the Teflon bags is sufficient to meet the experimental requirements, without the need for further treatment, rather than implying that any "passivation" step should been performed.

In line 105-110 of revised manuscript, "Firstly, selected branches were enclosed within a Teflon bag with a volume ranging from 15 to 60 L (Welch Fluorocarbon, Inc., USA) and PAR transparency close to 100%, which was passivated to avoid the generation and adsorption of VOCs as much as possible." is changed to "First, selected branches were enclosed within a Teflon bag (Welch Fluorocarbon, Inc., USA) with a volume ranging from 15 to 60 L and a PAR transparency close to 100%. The bag was made of polytetrafluoroethylene, a material known for its inherent chemical inertness that minimizes the generation and adsorption of VOCs without requiring additional treatment (Zhang et al., 2022).".

Reference:

Zeng, J., Zhang, Y., Zhang, H., Song, W., Wu, Z., and Wang, X.: Design and characterization of a semi-open dynamic chamber for measuring biogenic volatile organic compound (BVOC) emissions from plants, Atmos. Meas. Tech., 15(1), 79–93,

https://doi.org/10.5194/amt-15-79-2022, 2022.

9. L173. How do these 79 species compare? Consider adding an analysis or figure that enables the derivation of research conclusions.

**Response:**

Thank you for your insightful question and valuable comment. It is necessary to highlight the analysis of the 79 species with both dynamic and static measurements for a direct enclosure methodological comparison. In fact, we had already conducted a statistical analysis on this specific dataset, and the results were presented in the original manuscript in **Section 3.3.4. "Variability in emission rates derived from dynamic and static enclosure measurements"**. In that section, we concluded that "To conclude, the emission for plants with high intensity may be underestimated when measured by the static enclosure technique, while those for plants with low intensity may be overestimated." as described in line 380-382.

Furtherly, in order to make this comparison more accessible and enable clearer the research conclusions, we have now added a new scatter plot with linear regression as Figure 6 in the revised manuscript. This figure provides a direct visual comparison of the emission rates obtained from both enclosure techniques for these 79 species.

In line 372-373 of the revised manuscript of, "Among these plants, 66% had emission rate of 0.02–4.2 $\mu g\ g^{-1}\ h^{-1}$ (in lower emission intensity)." is changed to "Among these plants, 66% had an emission rate of 0.02–4.2 $\mu g\ g^{-1}\ h^{-1}$ (in lower emission intensity) (Figure 6).".

[Figure]

**Figure 6.** Comparison of emission rates derived from dynamic (R-value = 1) and static

(R-value = 2) enclosure measurements for isoprene (a) and monoterpenes (b). (The solid line represents the 1:1 relationship, and the Roman numerals on each subgraph represent the emission category.)

10. L192. Please clarify what is meant by regular in distribution.

**Response:**

Thank you for your valuable comment. The phrase "regular in distribution" is used to describe a key statistical phenomenon: while individual plant emission rates may appear random and unpredictable, the overall pattern formed by a large number of observations is stable and predictable. Specifically, "Regular in distribution" indicates that when all individual measurements are aggregated and visualized as a frequency distribution (as shown in Figure 2), the data collectively form a clear, non-random statistical pattern. The most common form of such regularity is a distribution that approximates a normal distribution. In such a pattern, most measurements cluster around a central mean value (the peak of the curve), while extremely high or low values occur less frequently (the tails of the distribution). We have revised the relevant sentence in the manuscript to clarify this point.

In line 201-204 of revised manuscript, "The plant emission rates were inconsistent, but regular in distribution, falling into different intensity levels." is changed to "Although individual plant emission rates were inconsistent, they exhibited a clear regularity in distribution, forming distinct intensity levels. Most measurements clustered around a mean value (the peak of the curve), revealing an underlying statistical structure despite individual variability.".

11. L200/Fig.1. Consider homogenizing the y-axis scales and grouping by species to improve visual clarity.

**Response:**

Thank you for your careful review and valuable suggestion. According to your comment, we have homogenized the y-axis scales across all subplots and grouping the figures by species to improve visual clarity.

[Figure]

**Figure 2.** Frequency distribution of BVOC emission rates (Frequency distribution of BVOC emission rates observed by dynamic (left column: a, c, e) and static enclosure techniques (right column: b, d). The Roman numerals on each subgraph represent the emission categories identified by the frequency distribution.)

12. L218 and generally. Consider using medians instead of means, as they are more representative for this type of dataset.

**Response:**

We sincerely thank the reviewer for raising this critical point regarding the use of means versus medians for datasets with large variability. We fully agree with the reviewer that robust statistical measures are essential in such cases. In our study, the observation of "large SDs relative to the mean" was precisely the motivation for us to design a more sophisticated statistical approach, rather than relying on the simple arithmetic mean of the raw data. We would like to clarify that our final "representative emission rate" is not the raw mean reported in Table S3. Instead, we implemented a two-step robust estimation procedure: We first calculated the 95% confidence interval (CI) of the mean for each category. This step identifies the most probable range for the true population mean. We then calculated the average only of the data points that fall within this 95% CI. This method effectively trims the distribution by

excluding potential extreme outliers in both the upper and lower tails before averaging. The resulting estimate is therefore a robust mean that is stabilized against the undue influence of extreme values, aligning with the core intention behind the reviewer's suggestion to use the median. Therefore, we believe that our current methodology already addresses the reviewer's concern and provides a statistically sound and representative estimate. We have revised the manuscript text to more explicitly describe this process as a "robust estimation" to avoid any potential misunderstanding.

In line 237-239 of revised manuscript, "Therefore, additional statistical analysis was undertaken to determine the representative emission rates, separately using all the values in each category which each had a normal distribution." is changed to "Therefore, to obtain a robust estimate of the central tendency that is less sensitive to potential outliers and the large observed variance, we implemented a two-step statistical protocol to determine the representative emission rates for each category.".

13. L249/Fig. 2. Please correct the English and refine the caption to improve clarity.

**Response:**

Thank you for your valuable comment. We are sorry for the poor English. In revised manuscript, we correct the English and refine the caption of Figure 3 in the revised manuscript to improve clarity.

[Figure]

**Figure 3.** Determination of the emission rate for a certain plant species. Step 1–4 mean the

order of priority when determining emission rates, namely using baseline data of the plant species primarily (step 1), then its belonging genus (step 2) and family (step 3), and at last all the plants in our dataset (step 4). For one of the steps, if no data fall within the interval of any emission category, then the observations of emission rates in the next step are used.

14. L277. This sentence ("Generally, most plant species performed low and moderate intensity of isoprene.") clearly illustrates how the English phrasing can make it difficult to understand the meaning of several statements in the paper.

**Response:**

Thank you for pointing out the unclear phrasing in this sentence and elsewhere in the manuscript. We are sorry for the awkward wording and we did not convey the scientific meaning clearly. We have thoroughly revised the sentence and carefully checked the entire manuscript to improve the clarity and precision of the English expression. The specific revisions in this paragraph are detailed below.

In line 293-298 of revised manuscript, "Generally, most plant species performed low and moderate intensity of isoprene. Specially, broadleaf plants mostly had a moderate emission intensity, while coniferous plants mostly had a lower intensity. For the monoterpene emission, both broadleaf and coniferous plants mostly had a moderate emission intensity. For herbs, the emission intensity of isoprene and monoterpenes varied greatly and covered low, moderate, and high levels." is changed to "In general, most plant species emit isoprene at low to moderate intensities. Specifically, broadleaf plants predominantly exhibited a moderate emission intensity, whereas coniferous plants were mostly characterized by low-intensity emissions. Regarding monoterpenes, both broadleaf and coniferous plants primarily showed a moderate emission intensity. In contrast, herbaceous plants displayed a wide range of emission intensities for both isoprene and monoterpenes, covering low, moderate, and high levels."

15. L305. Perhaps this recent review (https://doi.org/10.1038/s43247-023-01175-9) is more suitable for supporting this statement.

**Response:**

Thank you for your valuable suggestion. In line 321 of revised manuscript, the reference "(Tani et al., 2024; Yang et al., 2021)" is changed to "(Bourtsoukidis et al., 2024)".

Reference:

Bourtsoukidis, E., Pozzer, A., Williams, J., Makowski, D., Peñuelas, J., Matthaios, V. N., Lazoglou, G., Yañez-Serrano, A. M., Lelieveld, J., Ciais, P., Vrekoussis, M., Daskalakis, N., and Sciare, J.: High temperature sensitivity of monoterpene emissions from global vegetation, Commun. Earth Environ., 5, 23, https://doi.org/10.1038/s43247-023-01175-9, 2024.

16. L310/Fig.3. Consider removing the lines connecting the medians of each bar and removing the dashed box. Homogenizing the y-axes would also improve clarity.

**Response:**

Thank you for your valuable suggestion on the visualization of Figure 3. We have revised the figure according to all the suggestions to improve its clarity and consistency. The updated version is presented in the revised manuscript (Line 326, Figure 4).

[Figure]

**Figure 4.** Statistics of BVOC emission rates in various vegetation types. a–i: Distribution of emission rates for isoprene (a), monoterpenes (d), and sesquiterpenes (g) across vegetation types (trees, shrubs, herbs, and crops). Differences in BVOC emission rates between various subtypes of trees (b, e, h) and shrubs (c, f, i). Bar charts display median and mean of the distribution; bar ends represent the 25th and 75th percentiles, and outliers are also displayed.

17. L314. The text refers to boxplots, but Figure 3 shows bars. Please clarify.

**Response:**

Thank you for bringing this inconsistency to our attention. We sincerely apologize for this error in the figure caption. Figure 3 in the original manuscript uses bar charts to represent the data, not boxplots. We have now corrected the caption for Figure 4 in the revised

manuscript.

In line 327-331 of revised manuscript, "The boxplots display median and mean of the distribution, the ends of the boxes represent the 25th and 75th percentiles, and outliers are also displayed in the figure." is changed to "Bar charts display median and mean of the distribution; bar ends represent the 25th and 75th percentiles, and outliers are also displayed.".

18. L312. Consider citing a more general or representative publication. E.g. https://doi.org/10.1016/j.tplants.2009.12.005

**Response:**

Thank you for your valuable suggestion. We agree that the new reference could better support our statement. Accordingly, we have updated the citation. In line 336 of revised manuscript, "(Benjamin and Winer, 1998)" is changed to "(Peñuelas and Staudt, 2010)".

Reference:

Peñuelas, J., and Staudt, M.: BVOCs and global change, Trends in Plant Science, 15, 133–144, https://doi.org/10.1016/j.tplants.2009.12.005, 2010.

19. L343. The word anyway is typically used in informal communication and should be avoided in research writing. This is another example of the linguistic revisions needed throughout the manuscript.

**Response:**

Thank you for your valuable suggestion and correction. Here we revised "anyway" to "therefore" in the revised version. We have also taken this opportunity to conduct a thorough linguistic revision throughout the entire manuscript to ensure a formal and professional tone.

In line 358-360 of revised manuscript, "Anyway, to have more precise emission rate library, it is necessary to conduct more emission observations in the future to cover as many plant species as possible." is changed to "Therefore, expanding emission observations to cover a wider range of plant species is imperative for the development of a more precise emission rate library.".

20. L350/Fig. 4. The concept of this figure is good, but in practice it is difficult to follow all

the drawn lines.

**Response:**

Thank you for your valuable suggestion. we have taken the following steps to significantly improve the figure's readability. Firstly. we have revised the original Figure 4 by adjusting colors and line styles to enhance visual distinction and reduce clutter. Furthely, in order to mitigate the issue of overcrowding and improve the clarity, we have created three supplementary subfigures to provide isoprene, monoterpene and sesquiterpene emission categories of different genera in families Poaceae and Fabaceae, respectively (Figure S3).

In line 363-366 of revised manuscript, the colors and line styles of Figure 5 is adjusted.

[Figure]

**Figure 5.** Emission categories of the plant species in different genera of the families Poaceae and Fabaceae. Box length represents the number of species, and colors at the start and end of each connecting line correspond to the two connected ends.

In line 22-24 of revised supplementary information, Figure S3 is added.

[Figure]

**Figure S3.** Emission categories of the plant species in different genus of families Poaceae and Fabaceae for isoprene (a), monoterpenes (b), and sesquiterpenes (c).

21. L366-367. Please show this through a figure and strengthen the scientific interpretation.

**Response:**

Thank you for your valuable suggestion. In line 393-397 of revised manuscript, we have added a new scatter plot with linear regression as Figure 6 in the revised manuscript. This figure provides a direct visual comparison of the emission rates obtained from dynamic (R-value = 1) and static (R-value = 2) enclosure measurements for both isoprene and monoterpenes.

In line 372-373 of the revised manuscript, "Among these plants, 66% had emission rate of 0.02–4.2 µg g⁻¹ h⁻¹ (in lower emission intensity)." is changed to "Among these plants, 66% had an emission rate of 0.02–4.2 µg g⁻¹ h⁻¹ (in lower emission intensity) (Figure 6).".

[Figure]

**Figure 6.** Comparison of emission rates derived from dynamic (R-value = 1) and static (R-value = 2) enclosure measurements for isoprene (a) and monoterpenes (b). (The solid line represents the 1:1 relationship, and the Roman numerals on each subgraph represent the emission category.)

22. L382. These are not the only variables used in MEGAN. In addition, the meteorological parameters are not included directly in the MEGAN code but must be derived from external atmospheric data.

**Response:**

Thank you for your precise and expert correction regarding the MEGAN model setup. We apologize for the oversimplification and inaccuracy in our original description. We have now revised the sentence in the manuscript to accurately reflect the model's mechanics as suggested.

In line 414-417 of revised manuscript, "The variables driving MEGANv3.2 include vegetation data, meteorological parameters, and emission rates." is changed to "The simulation was driven by key inputs, including vegetation data, species-specific emission rates, and externally sourced meteorological fields such as Weather Research and Forecasting (WRF) output.".

23. L402. Please clarify how you updated plant species when MEGAN operates on plant functional types.

**Response:**

Thank you for this insightful question, which allows us to clarify a key aspect of our methodology. In earlier versions of MEGAN (such as 2.1) typically operate using PFT-averaged emission factors. In MEGAN version 3.2 that we used, both PFT distribution and detailed vegetation species composition in grids are inputted. Based on the vegetation composition, the gridded PFT-averaged emission factors can be calculated from the species-specific emission factors using the emission factor processing module of MEGANv3.2, and are then included in the emission calculator. Notably, the plant species in our library don't cover all the plants in the vegetation speciation file, for plants being not included in our library, we assign their emission factors by the global ones.

We apologize for the confusion caused by our initial wording. The manuscript has been revised to clearly emphasize this mapping methodology. In line 432-438, "The plant species-specific emission rates in the simulation were derived from our developed localized library." is changed to "In MEGAN version 3.2 used in this study, both PFT distribution and detailed vegetation species composition in grids are entered. Based on the vegetation composition, the gridded PFT-averaged emission factors can be calculated from the species-specific emission factors using the emission factor processing module of MEGANv3.2 and are then included in the emission calculator. Notably, plant species in our library did not cover all the plants in the vegetation speciation file; for species not included in our library, we assigned their emission factors using the global values.".

24. L410-418. This section is very hard to follow. Please rewrite and consider adding a table to improve clarity.

**Response:**

Thank you for this constructive suggestion. We are sorry for the unclear description of the simulation setups. Following your advice, we have now thoroughly revised the section to clarify the objectives, data sources, and priority rules of emission rates for each simulation experiments in a more logical and structured manner. And we have modified Table 1 to summarize the key design of all simulations to improve clarity.

In line 451-469 of revised manuscript, "Four simulations were established to explore the

impacts of quality of emission rates on BVOC emissions, as shown in Table 1. All the simulations had the same input other than emission rates. In Simulation 1, our developed emission rate library was fully applied including the emission rates with R-value of both 1 and 2, but those with R-value = 1 were selected preferentially, then supplemented by those with R-value = 2. This simulation made the emission rates to be localized as much as possible. Due to the higher accuracy of emission rates with R-value = 1 than those with R-value = 2, in Simulation 2, only those with R-value = 1 were applied, which would be expected to have more precise estimates. To investigate the impact of emission rates in different quality on the BVOC emission estimates, the results of Simulation 1 and 3, which was set by applying the emission rates with R-value = 2 preferentially, then supplemented by those with R-value = 1. Simulation 4 utilized the global library in MEGANv3.2. By comparing results in Simulation 1 and 4, the differences could be explored after localizing the emission rates." is changed to "To systematically evaluate the performance of the localized emission library developed in this study, four simulation experiments (Simulation 1–4) were conducted under identical model configurations, with the only variation being the source of emission rates. Simulation 1 incorporated all available species-specific emission rates in our localized library, prioritizing those with high accuracy (R-value = 1) and supplementing with lower-accuracy records (R-value = 2) where necessary, thereby maximizing localization. Simulation 2 used only R-value = 1 emission rates to establish a baseline under the most reliable data scenario. Simulation 3 also employed the full localized library but favored R-value = 2 data and supplemented it with R-value = 1. This design enabled a controlled assessment of data quality influence through comparison with Simulation 1. In Simulations 1–4, for plants without assigning localized emission rates, global data were used. Finally, Simulation 4 relied on the default global emission rate library embedded in MEGANv3.2, serving as a reference to quantify the net effect of emission rate localization when compared with Simulation 1. A summary of the simulation design is provided in Table 2 for clarity.".

**Table 2.** Simulation experiments for evaluation of the localized emission rate library.

| Simulations | Emission rate | | | Objective |
| | Local data with R-value =1 | Local data with R-value =2 | Global data in MEGANv3.2 | |
| --- | --- | --- | --- | --- |
| Simulation 1 | √√√ | √√ | √ | To maximize localization and represent the basic estimate using all available local data |
| Simulation 2 | √√√ | - | √√ | To provide an estimate based solely on the high-quality local data |
| Simulation 3 | √√ | √√√ | √ | To investigate the sensitivity of results to the data quality of emission rates, comparing with Simulation 1 |
| Simulation 4 | - | - | √√√ | To serve as a benchmark for quantifying the impact of localization |

Note: The number of √ symbols shows the priority from high to low. Taking Simulation 1 as an example, the plant species-specific emission rates are assigned from the localized library with R-value = 1 primarily (labeled √√√), then those with R-value = 2 as a supplement (labeled √√), and global data are used for plants without localized emission rates (labeled √).

25. L427. Butanes are primarily emitted from anthropogenic sources, with very minor contributions from vegetation. How can they emerge as the most important BVOCs? Such a statement reduces confidence in the scientific assessment and falls under General Comment 2.

**Response:**

Thank you for this constructive comment. We again checked the MEGAN simulation and indeed used the original output or butane emission for analysis in the paper. In our study, butane emission estimate applied the global emission factor without localization. It's emission factors range from 5 to 7.5 $\mu mol \cdot m^{-2} \cdot s^{-1}$, which are higher than the isoprene emission factor

from our localized library for some tree species, such as *Castanopsis Spach*, *Chamaecyparis formosensis*, and *Fagus hayata*. Their isoprene emission factor we used is 3 µmol m$^{-2}$ s$^{-1}$. Moreover, butane emission factors are higher than those of key monoterpenes we used for several species: for *Quercus acutissima*, the butane emission factor (0.006 nmol·m$^{-2}$·s$^{-1}$) exceeds that of α-pinene (0.005 nmol·m$^{-2}$·s$^{-1}$), while in *Picea asperata*, *Populus tremula*, and *Quercus rehderiana*, butane emission factors (0.0075, 0,006, 0.006 nmol·m$^{-2}$·s$^{-1}$) are greater than those of cis-β-ocimene (0.0004, 0.005, 0.0005 nmol·m$^{-2}$·s$^{-1}$). The usage of global emission factor likely introduced uncertainties and local observations are required.

In line 481-485 of revised manuscript, "Notably, in our study, the emission estimates for butane and isobutene used global emission factors without localization. They were even higher than the isoprene and monoterpene emission factors from our localized library for some tree species. This might have introduced uncertainties, and local observations for their emission rates are required in the future." is added.

26. L466-468. In such an anthropogenically dominated region, formaldehyde (HCHO) observations carry significant uncertainties that must be discussed.

**Response:**

Thank you for raising this critical point. We fully acknowledge that in an anthropogenically influenced region, a significant portion of the observed HCHO originates from anthropogenic precursors, which introduces uncertainty when using it as a proxy for biogenic emissions. Furthermore, we recognize that the satellite-retrieved HCHO VCD data itself carries inherent uncertainties.

In line 521-523 of revised manuscript, "To verify the performance in simulating the spatial distribution of emissions before and after applying the localized and global emission rates, the correlation between emission and observed formaldehyde (HCHO) vertical column density (VCD), was analyzed." is changed to "To evaluate the spatial patterns of simulated BVOC emissions based on various emission rates, the correlation between emissions and observed formaldehyde (HCHO) vertical column density (VCD) was analyzed.".

In line 527-531 of revised manuscript, "It is important to note that in our study region,

which is subject to anthropogenic influences, a substantial fraction of the atmospheric HCHO is expected to originate from anthropogenic VOCs (Ren et al., 2022). This likely caused uncertainty in our analysis, in summer. Meanwhile, satellite HCHO products also exhibit uncertainties (Chong et al., 2024)." is added.

References:

Ren, J., Guo, F., and Xie, S.: Diagnosing ozone–NOx–VOC sensitivity and revealing causes of ozone increases in China based on 2013–2021 satellite retrievals, Atmos. Chem. Phys., 22, 15035–15047, https://doi.org/10.5194/acp-22-15035-2022, 2022.

Chong, K., Wang, Y., Liu, C., Gao, Y., Boersma, K. F., Tang, J., and Wang, X.: Remote sensing measurements at a rural site in China: Implications for satellite NO2 and HCHO measurement uncertainty and emissions from fires, J. Geophys. Res.: Atmos., 129, e2023JD039310, https://doi.org/10.1029/2023JD039310, 2024.

27. L473-474 & L549-551. If I understand correctly, the values of R=1 together with R=2 provide the best result (simulation 1). If that is the case, how was the conclusion reached regarding important differences between static and dynamic measurements?

**Response:**

Thank you for your valuable comment, which allows us to clarify a key finding of our study. As you commented, simulation 1 yielded the best performance (correlation coefficient with HCHO VCD = 0.73). We can reach the conclusion regarding the implication of static and dynamic measurements used in the emission estimates through comparing results of Simulation 3 with Simulations 1 and 2.

(1) Comparison between Simulations 1 and 3: Both simulations were designed to maximize the use of localized emission rates, incorporating the same number of localized tree species, with the remaining species using global emission rates. The key difference lies in the priority given to data of different reliability levels: Simulation 1 prioritized the use of R-value = 1 data (dynamic measurements), while Simulation 3 prioritized R-value = 2 data (static measurements). As a result, Simulation 1 incorporated more R-value = 1 data than Simulation 3, whereas Simulation 3 used more R-value = 2 data than Simulation 1. The superior

performance of Simulation 1 (correlation coefficient, 0.73>0.63) indicates that greater application of dynamic measurement data leads to better model outcomes.

(2) Comparison between Simulations 2 and 3: Simulation 2 utilized R-value = 1 data primarily and supplemented the emission rates for other plants using global datasets. While Simulation 3 prioritized R-value = 2 data, supplemented by R-value = 1 data and then the global data. Aside from those commonly assigned using R-value = 1 or global data in both simulations, the comparison between the use of R-value = 1 data in Simulation 2 and R-value = 2 data in Simulation 3 further supports the same conclusion (correlation coefficient, 0.72 for Simulation 2 > 0.63 for Simulation 3).

Regarding to the important differences between static and dynamic measurements in line 549-551 in the original manuscript (line 612-613 in the revised manuscript), it is concluded by comparing emission rates from dynamic and static measurements in our localized library. As detailed in Section 3.3.4, a direct comparison of emission rates for the same plant species, measured by both static and dynamic methods, reveals a systematic bias. The results indicate that the emission for plants with high intensity may be underestimated when measuring by static enclosure technique, while that for plants with low intensity may be overestimated compared with the dynamic method. This pattern underscores a fundamental discrepancy between the two techniques.

We are sorry for the unclear description. We have now revised the description to clarify these questions in the revised manuscript. Line 584-592, "Their correlation coefficient was 0.63 (P < 0.05). Meanwhile, the correlation coefficient (0.72) for the emission estimated in Simulation 2 was also higher than that in Simulation 3. So, it can be concluded that the accuracy of the estimation decreased after introducing the emission rates with R-value = 2. Notably, compared to Simulation 2, Simulation 1 also introduced the measurements with R-value = 2, however, most of the species (84%) were herbs, and the coverage of them were limited. The above made the similar national total BVOC emissions and spatial accuracy of the two simulations." is modified to "Their correlation coefficient was 0.63 (P < 0.05), lower than that in Simulation 1. Meanwhile, the correlation coefficient (0.72) for the emissions

estimated in Simulation 2 was also higher than that in Simulation 3. Therefore, it can be concluded that greater application of emission rates from dynamic measurements leads to better implications for emission estimates. Together with maximizing localization, namely using emission rates from static measurements as a supplement, better results will be obtained. Notably, similar national total BVOC emissions and spatial accuracy were observed between Simulations 1 and 2 because most of the species (84%) with emission rates of R-value = 2 were herbs, whose coverage was limited.".

---

## Author Comment (AC2)

**Response to Referee #2:**

The submitted manuscript presents a new data library of species-specific BVOC emission rates in China. The authors describe the statistical method of assigning emission rates to individual plant species based on field measurements. They also provide the final dataset with emission rates of isoprene, monoterpenes and sesquiterpenes for 599 plant species. I consider these efforts essential for the improvement of regional BVOC emission modelling as they can reduce errors introduced when global databases are used. This is demonstrated in the results where the authors apply their emission rates to modelling with the MEGANv3.2 model and show improved spatial correlation of BVOC emissions in China with satellite observations when compared to the use of a global library. The methods are described in sufficient detail and the results support the presented conclusions.

**Response:**

Thank you for your positive and constructive comments. In this revision, we have strived to achieve the greatest improvement and hope you agree with our modification.

**Specific comments**

1. Section 3.1 on conversion of basal emissions - if there was a need to convert units of basal emissions from ug m$^{-2}$ h$^{-1}$ to ug g$^{-1}$ h$^{-1}$, could the authors please specify what values of specific or dry leaf matter (g/m$^2$) they used in conversion?

**Response:**

Thank you for your valuable suggestion. In this study, it was necessary to convert basal emission rates from a leaf area basis (µg m$^{-2}$ h$^{-1}$) to a leaf mass basis (µg g$^{-1}$ h$^{-1}$), the conversion was performed using the specific leaf area (SLA, in cm$^2$ g$^{-1}$) of the corresponding plant species. The SLA values applied in this conversion were not a single universal value but were instead compiled by individual plant species.

Firstly, for plant species whose emission rates were based on our own measurements, we utilized SLA values calculated by the general relationship between leaf area (LA) and leaf dry weight (LDW). Besides, for emission rates compiled from the literature, we first searched for the original publication. If the SLA value was provided therein, it was adopted directly to

maintain consistency with the reported emission data. If the source literature did not include SLA value, we conducted an extensive additional review of published studies to obtain a representative SLA value derived from measurements on the same species within China.

In line 158-164, "The specific leaf area (SLA) values used for conversion were species-specific. For our measurements, we utilized SLA values derived from the general relationship between leaf area and leaf dry weight. For literature-sourced emission rates, we preferentially used SLA values from the original publication when available; otherwise, we obtained representative SLA values from measurements of the same species in China through an extensive literature review (Ghirardo et al., 2016; Ren et al., 2014; Wang et al., 2017)." is added.

References:

Ghirardo, A., Xie, J., Zheng, X., Wang, Y., Grote, R., Block, K., Wildt, J., Mentel, T., Kiendler-Scharr, A., Hallquist, M., Butterbach-Bahl, K., and Schnitzler, J.-P.: Urban stress-induced biogenic VOC emissions and SOA-forming potentials in Beijing, Atmos. Chem. Phys., 16, 2901–2920, https://doi.org/10.5194/acp-16-2901-2016, 2016.

Ren, Y., Ge, Y., Gu, B., Min, Y., Tani, A., Chang, J.: Role of management strategies and environmental factors in determining the emissions of biogenic volatile organic compounds from urban greenspaces, Env. Sci. Technol, 48, 6237–6246, https://doi.org/10.1021/es4054434, 2014.

Wang, C., Zhou, J., Xiao, H., Liu, J., Wang, L.: Variations in leaf functional traits among plant species grouped by growth and leaf types in Zhenjiang, China, J. Forestry Res., 28 (2), 241–248, https://doi.org/10.1007/s11676-016-0290-6, 2017.

2. Section 3.3.1 is a bit hard to comprehend, it could be reduced and clarified or perhaps visualized.

**Response:**

Thank you for this constructive suggestion. As recommended, we have thoroughly revised Section 3.3.1 to enhance its clarity and readability. Also, Figure S2 is changed to show the composition of vegetation types in each emission category.

[revised manuscript text omitted]

In line 19-21 of revised supplementary information, Figure S2 is changed.

[Figure]

**Figure S2.** Composition of vegetation types and frequency statistics of plant species within each emission category of isoprene (a), monoterpenes (b), and sesquiterpenes (c).

3. In the modelled domain, what proportion of the area was covered with the emission rates with R-value 1?

**Response:**

Thank you for your insightful question. Following your comment, we have conducted a detailed analysis of the area coverage based on different vegetation types (trees, crops, shrubs, and herbs) rather than providing a summarized value, as we believe this offers a more accurate and detailed explanation. The key results are as follows.

For trees and crops, emission rates with R-value = 1 cover 93% of the total tree area and 94% of the total crop area in the domain. For shrubs and herbs, the coverage of R-value = 1 is lower, accounting for 34% and 21% of their respective coverage areas. This is a common challenge in regional BVOC modeling, as comprehensive field measurements for all shrub and herb species are often limited. Despite the scarcity of direct measurements of R-value = 1

for shrubs and herbs, their overall impact on the total BVOC emission is minor because of their low emission potential. This transparent assessment significantly enhances the confidence in the simulation results. Also, we have added a corresponding explanation in the manuscript to clear this point.

In line 444-450, "The application of emission rates with R-value = 1 was assessed by calculating the plant species coverage percentage of the total vegetation. Emission rates with R-value = 1 cover a high percentage of the dominant vegetation, specifically 93% of the total tree area and 94% of the total crop area. In contrast, their coverage is substantially lower for shrubs and herbs, with 34% and 21% of their respective areas. This is a common challenge in regional BVOC modeling, as comprehensive field measurements for all shrub and herb species are often limited." is added.

4. As the study also focuses on modelling with the MEGAN model, it would be interesting to see direct comparison of emission rates estimated in this study with those in the MEGANv3 global library (eg. by summarizing key differences) and/or with other available emission inventories.

**Response:**

Thank you for your valuable suggestion. Due to the unavailability of emission rates in other emission inventories, we compare our localized emission rate library and the MEGANv3.2 global one. Given that MEGANv3.2 often assigns identical data to all species within a genus, the representative emission rate (see Methods) for each genus was compared. Our analysis reveals both consistencies and notable discrepancies. For high-emission genera, the rank order is generally similar. However, the magnitudes often differ substantially. For isoprene, while genera like *Populus* and *Quercus* are high-emitters in both libraries, our localized emission rates for *Populus* (78.51 nmol m$^{-2}$ s$^{-1}$) and *Salix* (11.64 nmol m$^{-2}$ s$^{-1}$) differ from MEGAN's global value (37 and 37 nmol m$^{-2}$ s$^{-1}$, respectively). The contrast is even more pronounced for monoterpenes. Genera *Lespedeza* and *Spiraea* have the highest emission rates in both libraries, but our values (40.87 and 21.0 nmol m$^{-2}$ s$^{-1}$) are nearly an order of magnitude higher than the global value (5.30 and 2.73 nmol m$^{-2}$ s$^{-1}$). The emission rates of sesquiterpene

show closer agreement in both libraries. In summary, the key finding is that while the two libraries identify similar high-emitting genera, the MEGANv3.2 global emission rates are generally lower than our localized values.

In line 399-410 of revised manuscript, Section 3.3.5 is added as follows:

"**3.3.5 Comparison with global emission rate library of MEGANv3.2**

Comparison between our library and MEGANv3.2 global library was performed. For consistency, the comparison was conducted at the genus level, as the global library often assigns uniform values across species within a genus. Our results revealed consistent identification of high-emitting genera but quantitative differences (Figure S4). For isoprene, while genera like *Populus* and *Quercus* are high-emitters in both libraries, our localized emission rates for *Populus* (78.51 nmol m$^{-2}$ s$^{-1}$) and *Salix* (11.64 nmol m$^{-2}$ s$^{-1}$) differ significantly from the global value (37 and 37 nmol m$^{-2}$ s$^{-1}$, respectively). The discrepancies are even more pronounced for monoterpenes. Genera *Lespedeza* and *Spiraea* have the highest emissions in both libraries, but the localized values (40.87 and 21.0 nmol m$^{-2}$ s$^{-1}$) are nearly an order of magnitude higher than the global values (5.30 and 2.73 nmol m$^{-2}$ s$^{-1}$). In contrast, sesquiterpene emissions show closer agreement in both libraries."

In line 25-27 of revised supplementary information, Figure S4 is added.

[Figure]

Figure S4. Comparison of emission rates at the genus level between the MEGANv3.2 global library and the localized library developed in this study.

5. Section 4.2, the BVOC emission total in China is presented as 27.7 Tg / year 2020. Could the authors specify the unit, Tg of what? If different BVOC species (with different molecular masses) are to be summed, they need be converted to Tg of carbon before the summation. Please check and correct throughout the text. Same applies to Figs. 5 and 6. Also, if the BVOC split to percentage is to be presented as in Figure S4, first the units need to be harmonised to Tg (C), otherwise the masses are not comparable and the percentage is not valid. How are total BVOC emissions defined, which species are included in the sum?

**Response:**

Thank you for your critical comment and valuable suggestion regarding the units and summation of BVOC emissions. We agree that for the purpose of summing different BVOC species and calculating percentage contributions, harmonizing the units to a common metric is the scientifically ideal approach. In our study, the total BVOC emissions for China in 2020 was 27.7 Tg year$^{-1}$, representing the total mass of the emitted compounds (in Teragrams).

Accordingly, the unit of BVOC emissions is "Tg of compound per year". The unit we used mainly because the emissions of all BVOC compounds calculated by the MEGANv3.2 model (including isoprene, 40 monoterpenes, 45 sesquiterpenes, and 113 other VOCs) are measured in terms of mass, the final calculated total BVOC emissions are also based on the mass of the compounds, and we did not calculate them in terms of carbon. Actually, the BVOC emissions could be reported using either the total mass of the compound (Tg year$^{-1}$) or the mass of carbon (Tg C year$^{-1}$). Both units are valid and widely used (Wang et al., 2024).

In line 413-414 of the revised manuscript, "MEGANv3.2 was applied to estimate BVOC emissions." is changed to "MEGANv3.2 was applied to estimate BVOC emissions, including 199 compounds (isoprene, 40 monoterpenes, 45 sesquiterpenes, and 113 other VOCs).". In the caption of Figure S5, a note "All percentages are derived from emissions expressed in Tg compound." is added.

[Figure]

**Figure S5.** BVOC emission composition and top 10 compounds contributing the most to total emissions (BVOC emission composition of four categories. (isoprene, monoterpenes, sesquiterpenes, and other VOCs), and the top ten compounds of monoterpene, sesquiterpene, and other VOC categories (a); top 10 compounds and their belonging category contributing the most to total emissions(b). All percentages are derived from emissions expressed in Tg

compound.)

**Response:**

Thank you for your valuable suggestion. The newly designated Figure S7 in the supplementary information explicitly shows the correlation between spatial distribution of observed HCHO VCD and our simulated isoprene emissions across different model scenarios.

In line 37-40 of revised supplementary information, "Figure S7" is added. In the revised manuscript, line 531-533, "The isoprene emissions in July in Simulation 1 correlated stronger with HCHO concentration spatially (correlation coefficient = 0.73, P < 0.05) than Simulation 4 (correlation coefficient = 0.67, P < 0.05)." is changed to "As shown in Figure S7, the isoprene emissions in July in Simulation 1 correlated more strongly with HCHO concentration spatially (correlation coefficient = 0.73, P < 0.05) than in Simulation 4 (correlation coefficient = 0.67, P < 0.05).".

[Figure]

**Figure S7.** The correlation between isoprene emission and observed formaldehyde (HCHO)

vertical column density (VCD) in July 2020 during various simulations. (a–d: Simulation 1 (a), Simulation 2 (b), Simulation 3 (c), and Simulation 4 (d).)

9. Please double check for language and typing errors, some of which are pointed out below.

**Response:**

Thank you for your meticulous review and pointing out these language and typographical errors. We apologize for the oversight and we have carefully checked the entire manuscript and corrected all the specific errors listed, as well as other minor issues we identified during a thorough proofreading. The changes made to each point are detailed below. Besides, English throughout the manuscript is edited by Elsevier Language Editing Services. The certificate is as follows. The changes caused by the revision of English are not marked in revised manuscript, and are not listed here due to the limited space.

[Figure]

**Certificate of Elsevier Language Editing Services**

**The following article was edited by Elsevier Language Editing Services:**

A localized plant species-specific BVOC emission rate library
of China established using a developed statistical approach
based on field measurements

**Ordered by:**

Lingyu Li

**Estimated Delivery date:**
2025-12-04
**Order reference:**
ASLESTD1120118

[Figure]

- Line 19: pose -> poses

**Response:**

Thank you for your valuable suggestion. In line 18 of revised manuscript, "pose" is changed to "poses".

- Line 20: exited -> existed

**Response:**

Thank you for your careful review and valuable suggestion. We are sorry for this spelling error. After edited by Elsevier Language Editing Services, "exited" is changed to "exists" in line 19 of revised manuscript.

- Line 104: on -> of

**Response:**

Thank you for your valuable suggestion. In line 99 of revised manuscript, "on" is changed to "of".

- Line 151: environment -> environmental

**Response:**

Thank you for your valuable suggestion. In line 155 of revised manuscript, "environment" is changed to "environmental".

- Line 162: remove space after )

**Response:**

Thank you for your valuable suggestion. In line 167 of revised manuscript, the space after ")" is removed.

- Line 200: rates(a-e -> rates (a-e

**Response:**

Thank you for your valuable suggestion. In line 213-214 of revised manuscript, "rates(a-e" is changed to "emission rates observed by dynamic (left column: a, c, e) and static enclosure techniques (right column: b, d).".

- Line 254: were -> was

**Response:**

Thank you for your valuable suggestion. In line 278 of revised manuscript, "were" is changed to "was".

- Sentence on lines 413-416 is somewhat unclear and possibly missing a verb

**Response:**

Thank you for your valuable suggestion. We are sorry for the unclear description of the simulation setups. Following your advice, we have added a verb and rewritten the sentence to clarify the objectives.

In line 458-460 of revised manuscript, "To investigate the impact of emission rates in different quality on the BVOC emission estimates, the results of Simulation 1 and 3, which was set by applying the emission rates with R-value = 2 preferentially, then supplemented by those with R-value = 1." is changed to "Simulation 3 also employed the full localized library but favored R-value = 2 data and supplemented it with R-value = 1. This design enabled a controlled assessment of data quality influence through comparison with Simulation 1."

- Line 461: less -> lower

**Response:**

Thank you for your valuable suggestion. In line 516 of revised manuscript, "less" is changed to "lower".

- Line 524: emission -> emissions

**Response:**

Thank you for your valuable suggestion. In line 583 of revised manuscript, "emission" is changed to "emissions".

- Sentence on lines 528-529 is somewhat unclear.

**Response:**

Thank you for your valuable suggestion. We agree that the original description of the simulation setups was unclear. Following your advice, we have now completely rewritten the section as follows.

In line 589-592 of revised manuscript, "The above made the similar national total BVOC emissions and spatial accuracy of the two simulations." is changed to "Notably, similar national total BVOC emissions and spatial accuracy were observed between Simulations 1 and 2 because most of the species (84%) with emission rates of R-value = 2 were herbs, whose coverage was limited.".

- Sentence on lines 547-549 - missing verb.

**Response:**

Thank you for your valuable suggestion. In line 608-610 of revised manuscript, "Based on this, a localized plant species-specific BVOC emission rate library for China, including isoprene, monoterpene, and sesquiterpene emission rates for 599 plant species." is changed to "Based on this, a localized plant species–specific BVOC emission rate library for China was developed, including isoprene, monoterpene, and sesquiterpene emission rates for 599 plant species.".